# Mature fault mechanics revealed by the highly efficient 2025 Mandalay earthquake

Eric O. Lindsey [1], Yu-Ting Kuo [2], Yu Wang [3] ✉, Myo Thant [4,5,6] & Tha Zin Htet Tin [5,6]

Understanding the causes of variability in earthquake rupture characteristics, particularly the distribution of shallow slip and the extent of off-fault deformation, is crucial for refining seismic hazard models and understanding fundamental rupture mechanics. The 28 March 2025 magnitude 7.7 earthquake on Myanmar's Sagaing fault, the largest instrumentally recorded event in the region, is an extreme example of rupture behavior on mature, geometrically simple strike-slip systems. Here, we use Interferometric Synthetic Aperture Radar (InSAR) and optical satellite imagery to characterize the coseismic surface deformation field and infer the slip distribution along its ~500-km rupture. Our results reveal a remarkably sharp surface rupture with effectively zero shallow slip deficit along its entire length; peak slip consistently occurred at or near the surface, contrasting sharply with recent large earthquakes on less mature faults with significant off-fault deformation and shallow slip deficit. Furthermore, while the rupture smoothly propagated through seismic gaps, we show that it also re-ruptured segments previously broken in 1929, 1930 and 1956, but with reduced slip in these zones – consistent with a slip-predictable model. These observations highlight that rupture processes on mature plate boundary faults can be highly efficient, with implications for seismic hazard assessment on similar strike-slip fault systems worldwide.

Accurately characterizing coseismic slip near the Earth's surface is critical for both seismic hazard assessment and the interpretation of geologic fault slip histories. Models of near-fault ground shaking are highly sensitive to both the magnitude and depth of slip, while geologic estimates of long-term slip rates often rely on measuring cumulative offsets at the surface[1–3]. However, geodetic observations from numerous large strike-slip earthquakes have suggested that slip inferred at seismogenic depths commonly does not fully propagate to the surface on the main fault trace[4–7]. This phenomenon, termed the Shallow Slip Deficit (SSD), implies potential underestimation of both near-fault shaking in hazard models and cumulative displacements in geologic slip rate studies[8–10].

SSD is often attributed to shallow inelastic processes and distributed slip on secondary structures – collectively termed off-fault deformation (OFD)[11–13]. High-resolution imagery and field studies have increasingly quantified significant OFD, demonstrating that it can account for tens of percent of the total slip budget, particularly along geometrically complex or immature faults[7,13–15]. Consequently, geologic offsets measured solely on the principal fault strand may systematically under-record the full deformation[16,17]. While fault maturity and geometric simplicity are often hypothesized to promote focused slip and reduce OFD, the degree to which they eliminate SSD remains debated, with some field and geodetic studies suggesting SSDs can persist even on relatively mature structures[18–20]. This highlights an

[1]Department of Earth and Planetary Sciences, The University of New Mexico, Albuquerque, NM, USA. [2]Department of Earth and Environmental Sciences, National Chung Cheng University, Chiayi, Taiwan. [3]Department of Geosciences, National Taiwan University, Taipei, Taiwan. [4]Myanmar Institute of Earth and Planetary Sciences, Yangon, Myanmar. [5]Myanmar Earthquake Committee, Yangon, Myanmar. [6]Department of Geology, University of Yangon, Yangon, Myanmar. ✉e-mail: wangyu79@ntu.edu.tw

incomplete understanding of the factors controlling shallow rupture mechanics.

Investigating these controls requires detailed studies of ruptures on diverse and well-characterized faults. The Mw 7.7 Mandalay earthquake is an excellent candidate rupture on a highly mature fault. The event struck central Myanmar on March 28th, 2025, rupturing the central part of the right-lateral Sagaing fault, which slips at a long-term rate of 20–24 mm/yr[21–27], accommodating about half of the northeastward motion between the India and Sunda Plates[28–30]. The fault is highly linear and has a generally simple surface expression[31], with a total offset of more than 200 km[32,33]. Similarities between the Sagaing fault, the San Andreas Fault in California, the Alpine fault in New Zealand[34], the Motagua fault in Guatemala[35], and the North Anatolian Fault in Turkey[36,37], among others, means the 2025 event provides a unique opportunity to study the rupture and deformation processes associated with large continental strike-slip earthquakes on mature faults, with implications for improved hazard estimates for nearby cities like San Francisco, Istanbul, and Guatemala City.

Eyewitness accounts and instrumental catalogs have recorded nine large (M > 6.5) earthquakes along the Sagaing fault over the past century[2]. Based on a combination of historical earthquake intensity records and along-strike changes in geomorphic expression of the fault, previous studies subdivided the central and southern Sagaing fault into five named sections[32,38], each of which is capable of generating M > 7 earthquakes every ~100 to ~300 years[2,27,39], or rupturing together as in 2025. Following significant earthquake clusters in the first half of the 20th century, including three large events near Bago in 1929–1930 and two north of Mandalay in 1946 (Fig. 1), the central section between the Sagaing ridge near Mandalay and a trans-tensional stepover near Naypyitaw (~180–250 km in length) was recognized as a remaining seismic gap, hypothesized to have last slipped in a major event during the destructive 1839 Ava earthquake[2]. Partially rupturing this gap was the 1956 Sagaing earthquake (Mw ~6.8), which was inferred to have slipped south of latitude 22°N[38]. More recently, the 2012 Mw 6.8 Thabeikkyin earthquake occurred at the northern margin of the gap, overlapping the 1946 Mw 7.7 rupture zone[32].

The prolonged quiescence since 1839 along the central segment (apart from the 1956 event) and evidence of deep interseismic locking[22–25] underscored a high potential for future large earthquakes[32]. However, the 28 March 2025 Mw 7.7 earthquake exceeded expectations by rupturing a much longer ~500 km central portion of the Sagaing fault, including areas that slipped during the 1956, 1929 and northern 1930 events (Fig. 1). The source-time function from the U.S. Geological Survey suggests that most of the energy was released during the first 60 seconds, with an average rupture velocity of ~4 km/s[40]. Recent seismological studies have confirmed the supershear nature of the rupture[41–43], in particular the southward-propagating portion of the rupture north of Naypyitaw. The northward rupture speed is slower, characterizing a subshear rupture between the earthquake epicenter and its northern termination. A large portion of the aftershocks were distributed between the epicenter and the rupture's northern end, with fewer events along the southern 400 km of the rupture[40,44] (Fig. 1). This could be indicative of either a relatively simple fault without significant heterogeneities or off-fault planes of weakness, or a smooth supershear rupture in this segment with relatively uniform stress drop[45,46].

Here, we quantify the fault slip during this earthquake based on sub-pixel tracking of optical and radar satellite imagery and interferometric synthetic aperture radar (InSAR). The earthquake ruptured multiple sections that had slipped at different times over the past two centuries, and we show that the co-seismic slip largely followed a slip-predictable model, with lower slip along the rupture zones of the 1930 and 1956 earthquakes. We also show that there is effectively zero SSD along the entire rupture: this earthquake represents a unique end-member example of highly efficient slip (i.e., slip concentrated on the fault with minimal off-fault inelastic deformation) during a major continental strike-slip earthquake. These results suggest that mature sections of other continental strike-slip faults worldwide may likewise be able to link up in unusually large ruptures in the future.

## Results and Discussion

### Large offsets, linked segments, and loose slip predictability

Due to ongoing conflict in the central part of Myanmar and widespread damage to the country's communication and transportation infrastructure caused by the earthquake, limited near-field observations were available immediately following the event. Hence, we used sub-pixel correlation of pre- and post-quake optical satellite imagery from Copernicus Sentinel-2C (02, 05 March 2025 and 01, 04 April 2025), and amplitude sub-pixel correlation and interferograms derived from the C-band Sentinel-1 SAR mission to constrain the surface offsets and coseismic deformation field along the Sagaing fault (see Methods). The sub-pixel correlation results from both the Sentinel-2 and the Sentinel-1 data indicate a very sharp pattern of displacement across the fault, revealing an extremely simple and continuous surface rupture extending ~510 km along the fault, bounded by the 2012 Thabeikkyin rupture zone to the north near 22.6°N and by a comparatively sharp bend in the fault at 18°N to the south (Fig. 1 and Fig. 2). This ultra-long rupture exceeds the known length of other continental strike-slip earthquakes mapped over more than a century, including the 1906 San Francisco earthquake[47], the 2002 Denali earthquake[20], the 2001 Kokoxili earthquake[48], and the 2013 Balochistan earthquake[16].

We mapped the surface offsets with a series of profiles across the displacement field (Fig. 3); the largest offsets exceed 6 m along the Sagaing Ridge, just north of the earthquake epicenter (Fig. 3b). The surface rupture south of the epicenter is much longer, propagating to nearly 18°N. Results derived from both the Sentinel-2 and the Sentinel-1 data show a sharp and simple arcuate fault trace without any notable bends or step-overs (Fig. 2). The average fault offset at the surface is approximately 4 m along the entire southward section, except for the area between Sagaing City and the Mandalay International Airport (MDL) and the segment south of 19°N, where the surface displacement drops to 3 m or less (Fig. 3b).

We constructed a finite slip model after carefully resampling the data to maintain accurate values close to the fault (see Methods). Prior interseismic observations near the Sagaing ridge had suggested variable eastward fault dip as shallow as 70 degrees[22–24,49] and a possible westward dip to the south near Bago[22]. We conducted a grid search for two dip angles in the north and south of the rupture, with a smooth variation between them. The best-fitting model has an eastward dip of 65° in the north and a westward dip of 80° in the south, and fits the data significantly better than a vertical fault model, although the slip distributions, peak slip, and overall moment release are nearly unchanged (Supplementary Figs. S7–S9). The eastward dip in the north agrees well with the interseismic observations, and while significantly off-vertical, is not the most extreme example of a strongly dipping strike-slip fault[50]. We also considered a model with two components of slip but found that a model with strike slip only is preferred by an F-test; the two models are highly visually similar (Supplementary Fig. S11).

Our preferred slip model (Fig. 4) has a peak right-lateral strike-slip of 6.0 meters, and a total seismic moment of $5.29 \times 10^{20}$ N·m, corresponding to a moment magnitude ($M_W$) of 7.75, and an average shear stress drop across the ruptured portion of the fault of 3.9 MPa. Overall, our model suggests an exceptionally simple, shallow, and consistent rupture along the Sagaing Fault, with the depth extent gradually decreasing at the southern end. Most of the coseismic fault slip occurred on the fault interface between 0 and 10 km deep, with an average coseismic slip of ~4 m in the top 2–3 km. The maximum slip of 6 m occurred just north of the epicenter, slightly less than the offset measurements (Fig. 3) but consistent with a rupture having peak slip at

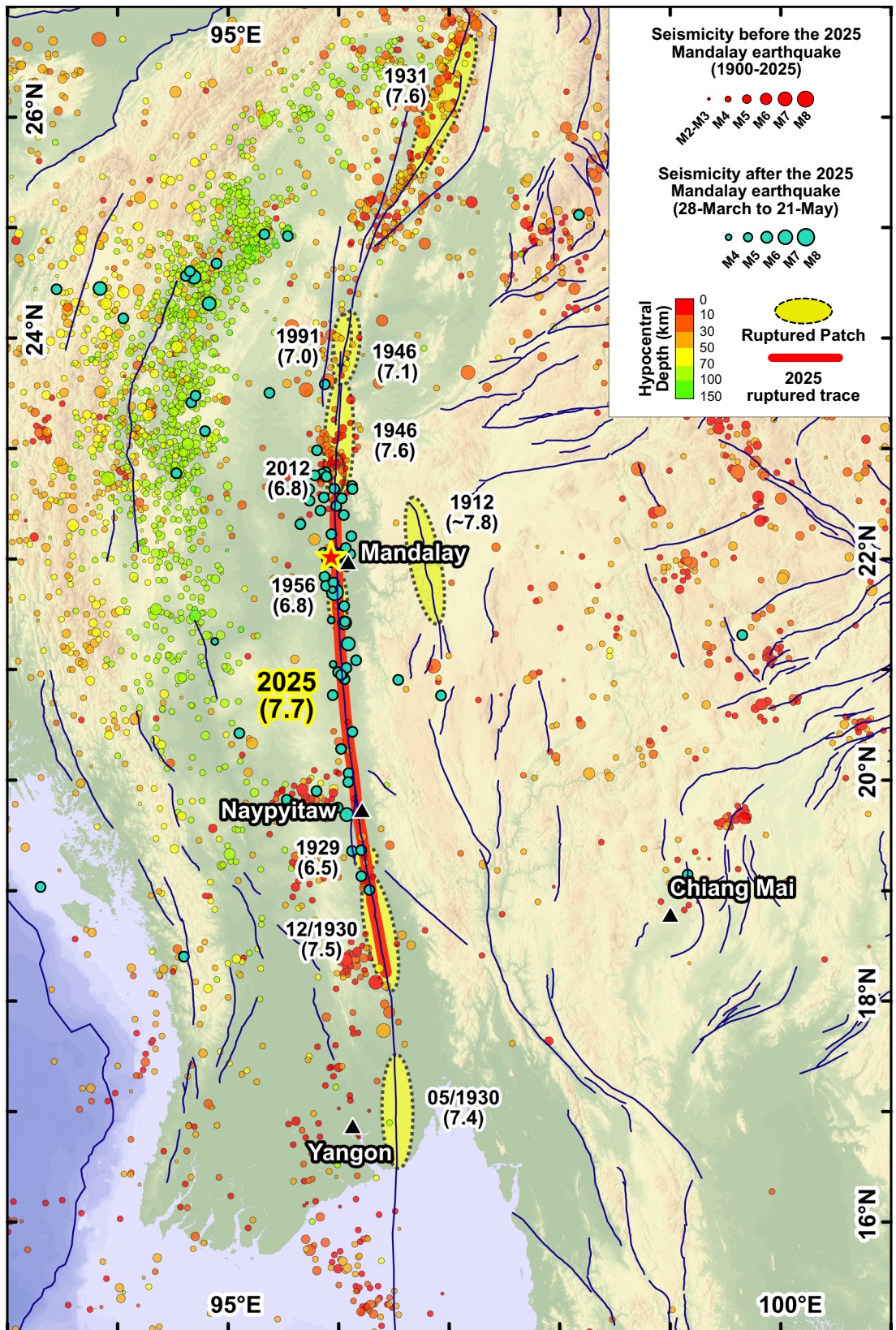

**Fig. 1 | Location of major historical earthquakes and the 2025 Mw 7.7 Mandalay earthquake along the Sagaing fault.** Active Tectonics and earthquakes along the Sagaing fault, highlighting the 2025 rupture patch[86] (red line) and previous inferred rupture zones (yellow ovals). The 2025 event extends ~500 km along-strike and fully overlaps the inferred 1956, 1929, and Dec 1930 earthquake rupture patches. Most of the aftershocks are located at the northern and southern ruptured sections, with fewer aftershocks distributed in the section north of Naypyitaw. Background seismicity (circles; 1900 to 27 March 2025) and earthquakes after the 2025 mainshock (outlined dark-cyan circles; 28 March 2025 to 21 May 2025) are from U.S. Geological Survey[40]. Active fault lines (blue) and fault ruptured patches are modified from ref. 5. Background topography derived from Shuttle Radar Topography Mission data[75,76]. Figure created using ESRI ArcGIS software.

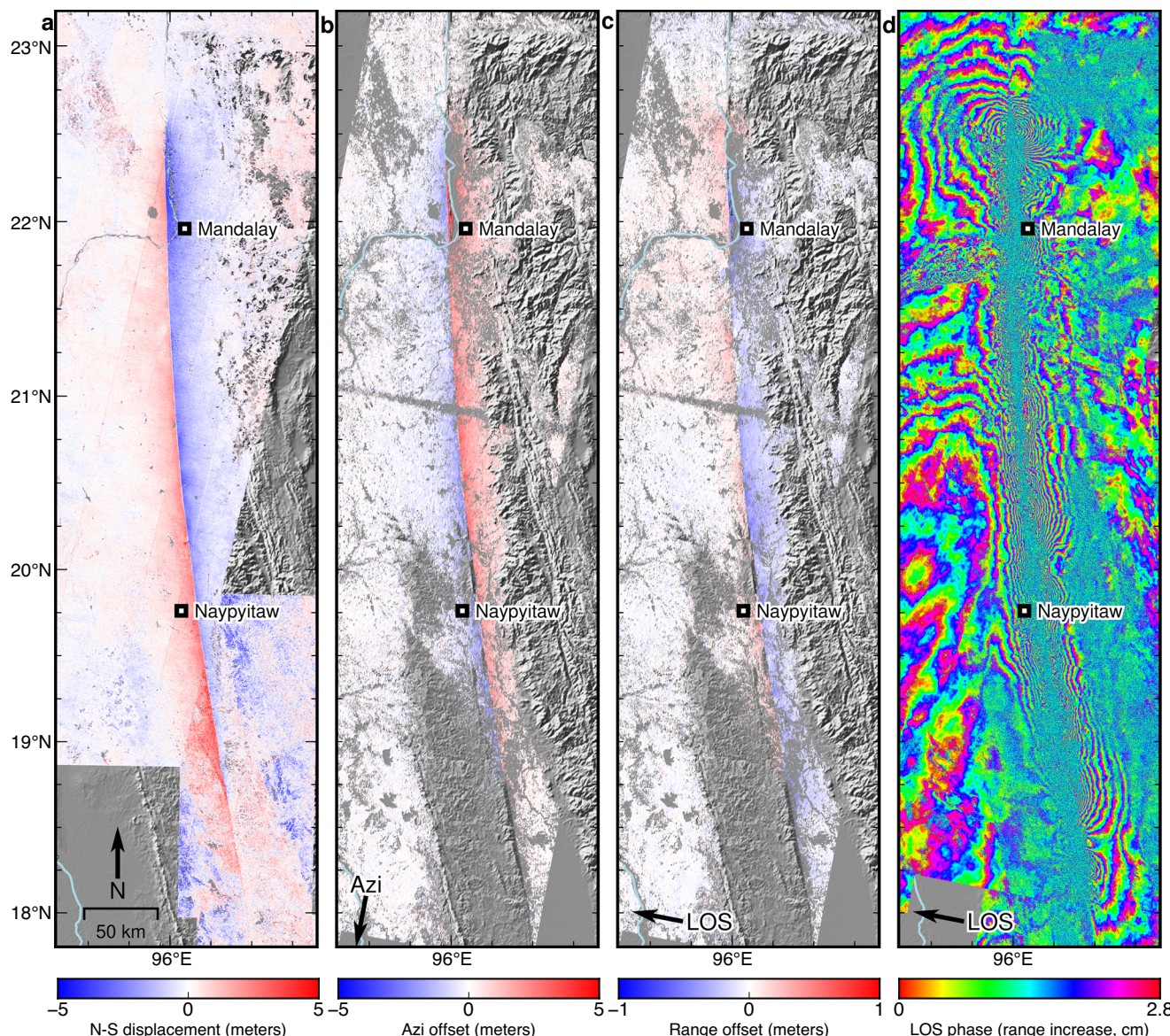

**Fig. 2 | Satellite-based ground deformation data used in this study.** Surface deformation derived from **a** the Sentinel-2 optical satellite imagery North-South component of motion (pre-event images 2025/03/02 and 2025/03/20, post-event images 2025/03/30 and 2025/04/01), **b** combined Sentinel-1 SAR image azimuth (Azi.) offsets from descending track 33 (2025/03/19 – 2025/03/31) and track 106 (2025/03/24 – 2025/04/05), **c** combined line-of-sight (LOS) range offsets from the same tracks, and **d** combined interferograms. All datasets are shown individually in the supplementary materials. Figure created using Generic Mapping Tools software[87] with hillshade derived from Shuttle Radar Topography Mission data[75,76]. Contains modified Copernicus Sentinel data [2025].

the surface. Such a large amount of slip could be related to shallow dynamic overshoot on the fault plane[51], or release of localized residual strain resulting from a potential low slip area in the previous earthquake event in 1839. A low slip area between 21.2°N–21.8°N coincides with the inferred 1956 rupture zone[38]. The modeled peak fault slip again exceeds 5 m near 20.2°N, in good agreement with the surface offset measurements. The rupture shallows further south of 19°N, with total slip reduced to just 2 m, until the main rupture ends between 18° and 18.2°N, as seen from the subpixel correlation results and SAR interferometry (Fig. 2).

The along-strike change in the rupture depth and the termination of the fault rupture match well with the known earthquake rupture history along the Sagaing Fault (Fig. 1). The low-slip patch between 22° and 21°N overlaps the inferred rupture zone of the M~7 Sagaing earthquake of 1956, which was inferred to have ruptured southward from 22°N[38]. The reduced slip south of 19°N corresponds to the rupture area of the December 1930 Pyu earthquake, which is constrained

by earthquake damage records and macroseismic intensity[52], with an inferred high macroseismic intensity zone matching well with the lowest slip region in the 2025 event (Fig. 4). The southern termination of the main rupture at a comparatively sharp bend in the fault is also the location where the Dec-1930 earthquake was inferred to terminate, indicating a possible persistent rupture barrier. Finally, the base of the seismogenic zone aligns well with the depths of relocated background seismicity; in particular, a small zone of high activity surrounding the 2025 epicenter, suggestive of a high pre-stress along that portion of the fault[53].

The qualitative match between the 2025 slip pattern, the estimated interseismic locking depth, and the inferred history of seismic slip along the fault suggests this long, multi-segment rupture largely released the available stress at any point on the fault, corresponding to a loosely slip-predictable model[3,54]. However, the 2025 coseismic slip within the 1929 and 1956 rupture patches is larger than expected given the time elapsed since the last event, suggesting that these two

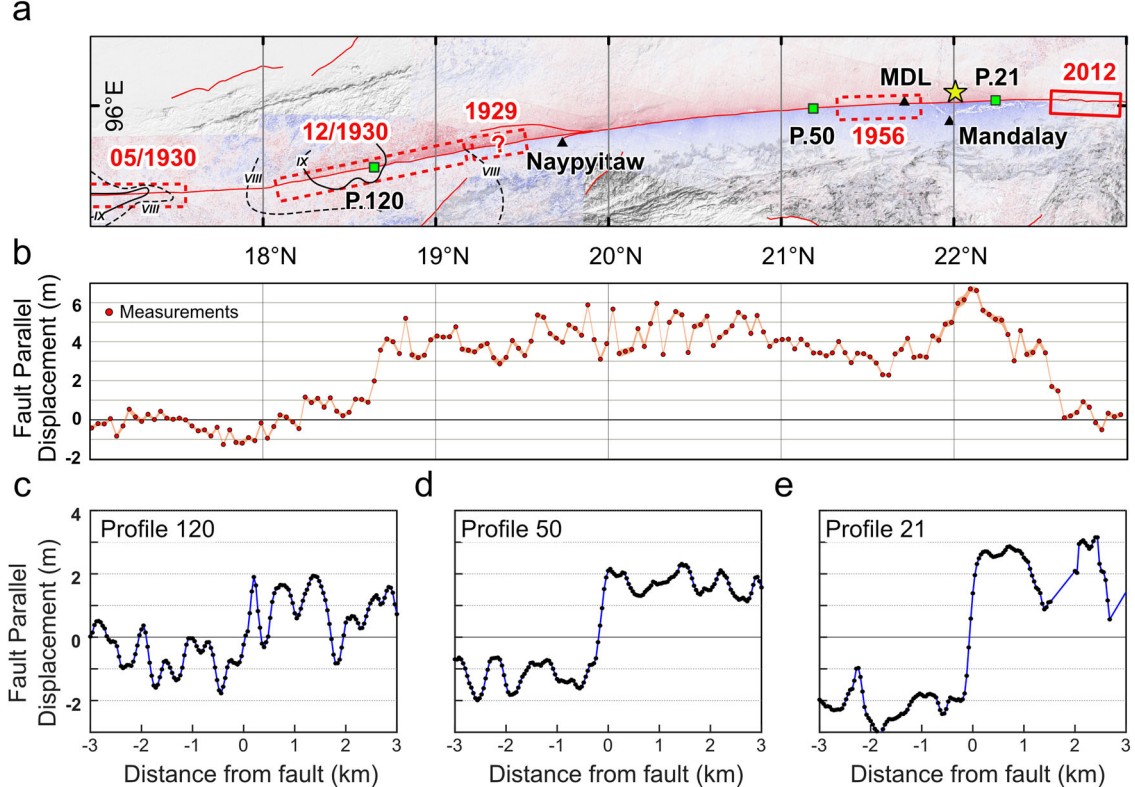

**Fig. 3 | Surface rupture and ground deformation associated with the 2025 Mandalay earthquake based on Sentinel-2 observations. a** Surface rupture and ground deformation derived from the subpixel-correlation result from the Sentinel-2 data. The solid (Intensity *IX*) and dashed (Intensity *VIII*) black lines are the isoseismal lines of the 1930 May and the 1930 December earthquakes, under the Rossi-Forel intensity scale. These Intensity data of both 1930 earthquakes were from ref. 27. Other earthquakes' rupture patches are modified from ref. 5. MDL, Mandalay International Airport **b** The fault-parallel motion across the fault along the ruptured section, estimated from 177 cross-fault profiles (see Methods), numbered from North to South. **c** shows the pattern of displacement across profile 120, within the inferred Dec. 1930 rupture area, **d** shows profile 50, just south of the 1956 rupture area, and **e** shows profile 21 near Mandalay, near the area of peak slip. The profile width is 440 m in these three profiles. The analysis of fault-parallel motions was performed in the software ENVI and Cosi-Corr. Figure created using ESRI ArcGIS software. Contains modified Copernicus Sentinel Data [2025].

previous events may not have completely released the stress on the interface. Comparatively, there is a significant decrease in coseismic slip during the 2025 event within the reported high-intensity zone[52] of the December 1930 earthquake (Fig. 4). The amount of slip in this zone (~2 m) matches the predicted slip deficit since the Dec-1930 earthquake, suggesting low residual stress on the core rupture patch of the Dec-1930 earthquake.

### Near-total absence of SSD

The concept of SSD emerged from early satellite geodetic studies of events like the 1992 Landers and 1999 Hector Mine earthquakes in California, where kinematic models suggested peak slip occurred several kilometers below the surface[4]. This kilometer-scale SSD is distinguished from OFD in the top tens of meters, which is associated with near-surface soil failure and does not indicate a decrease in total slip across the fault. Proposed explanations for the kilometer-scale SSD have included inherent rock properties near the surface (e.g., velocity-strengthening friction, or low normal stress promoting dilatancy) or the accommodation of strain via significant inelastic OFD within a much broader damage zone[11,12]. While debate continues regarding the influence of biased near-fault data, which can create modeling artifacts mimicking SSD[5,6,55], virtually all recent well-observed earthquakes exhibit at least some SSD, particularly along geometrically complex and less mature fault segments[7,13,15,17].

In the present case, our results indicate that slip propagated efficiently to the surface nearly everywhere along this exceptionally linear portion of the Sagaing fault. This near-total lack of SSD signifies an unusually efficient rupture process, where nearly all strain release was accommodated by slip on the main fault plane, with minimal energy dissipation through OFD, except perhaps within the top ~100–200 m, the spatial resolution limit of our method[56–60]. This aligns with the hypothesis that highly mature faults, characterized by long cumulative offsets and simple through-going structures, might facilitate highly localized slip, channeling rupture energy along the primary fault plane up to the free surface. We compare a large set of normalized slip profiles from recent large events to this event in Fig. 5. We find that rupture velocity (supershear or subshear) does not have a clear effect on the fault slip distribution with depth, as cases from both types of rupture showed various degrees of SSD (Figs. 5a, b). This is supported by the rupture in the 2025 Mandalay earthquake, in which both the northward propagating subshear rupture, and the southward propagating supershear rupture[41–43] show similar slip distributions with minimal SSD (Fig. 4). Figure 5c compares the estimated SSD for each of these events to the corresponding cumulative offset of each fault. Although previous studies suggest a wide range of SSD values and a complex cause of the SSD, particularly in smaller (M < 7) events[7,15], we find a strong qualitative inverse correlation between SSD and fault maturity in large (M > 7) events, with the Mandalay earthquake at the extreme end of the spectrum.

While highly mature faults tend to develop a simple and smooth fault trace, favoring the occurrence of large earthquakes with minimal SSD, the 2013 Mw 7.7 Balochistan earthquake, a partially supershear event in Pakistan complicates a simple interpretation based solely on fault maturity. As documented by Lauer et al. [50] this rupture exhibited

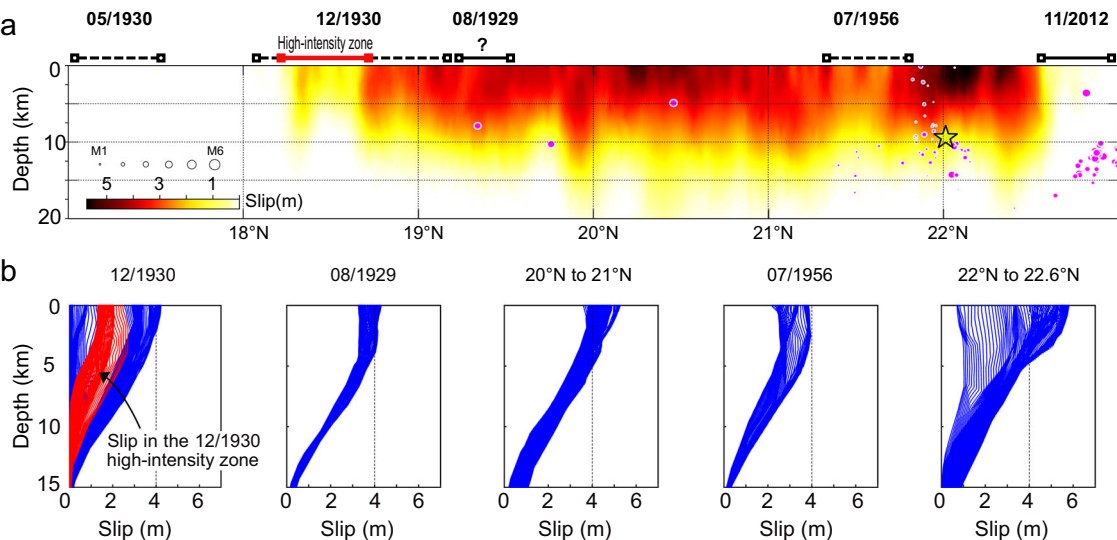

**Fig. 4 | Slip Distribution of the 2025 Mw 7.7 Mandalay earthquake. a** Modeled right-lateral fault slip distribution during the 2025 Mandalay earthquake. Black lines indicate the inferred extent of previous magnitude 6 or greater events in the past century. Purple colored circles are relocated earthquakes from Mon et al. 2020 and Fadil et al., 2023, showing a cluster near the hypocenter of the 2025 event. **b** shows a series of vertical slip profiles in blue within the indicated along-strike regions for each sub-panel. Profiles in red are from the zone reported to have experienced high-intensity ground shaking during the December 1930 event[52].

minimal SSD along most of its length, despite occurring on an immature fault with an estimated 11 km total strike-slip offset[61]. Notably, like the Mandalay event, the Balochistan rupture trace is remarkably simple, suggesting fault linearity might be the key for efficient slip propagation to the surface on the main fault; while mature faults tend to be more linear and simple, reactivation of other pre-existing structures can also create highly linear faults that rupture with low SSD[50]. High-resolution mapping by Gold et al.[16] quantified significant OFD accommodating ~28% of the slip budget across a shear zone with a width of ~10 m –1 km in that event. Our results suggest that the 2025 Mandalay event also experienced some degree of OFD within tens of meters of the fault in some locations (Fig. 3e). Both cases highlight that a minimal SSD does not necessarily preclude substantial energy dissipation via OFD at small scales surrounding the fault trace.

The magnitude of the observed surface slip in the 2025 Mandalay event presents an additional puzzle. Peak surface offsets greater than 6 m (Figs. 3, 4) exceed the 3.7 to 4.4 m of slip deficit hypothesized to have accumulated since the last inferred major rupture in 1839, given a slip rate of 20–24 mm/yr[21–27]. This apparent shallow slip 'excess' relative to the simple post-1839 deficit calculation could potentially represent dynamic overshoot, where rupture dynamics cause transient slip to exceed the stored elastic strain; this has been inferred in subduction zone settings[62]. On the other hand, it might indicate the 2025 event released strain deficits accumulated over multiple earthquake cycles, perhaps because the 1839 event ruptured incompletely. Distinguishing between these possibilities is critical: dynamic overshoot implies potentially higher-than-expected near-fault ground shaking[51,63,64], with impacts for hazard assessments, while multi-cycle deficit release would challenge assumptions about characteristic slip in both co- and paleoseismic studies and long-term fault behavior models[3,18,65].

The 2025 Mandalay earthquake provides a benchmark for rupture behavior on mature, geometrically simple strike-slip faults. Its ~500 km length, remarkable surface-rupturing efficiency with negligible SSD, and ability to link segments with varying seismic histories challenge paradigms based on less mature systems. Observations from the 2025 Mandalay earthquake, along with observations from events including the 1999 Izmit, the 2001 Kokoxili, and the 2002 Denali earthquake[66], strongly suggest that similarly efficient, long, and potentially multi-segment ruptures should be anticipated along linear, mature sections of other major plate boundaries, such as the San Andreas fault. Recognizing this potential for highly focused, surface-reaching slip is crucial for refining seismic hazard assessments and preparing for future large earthquakes on these key plate boundary systems.

## Methods
### Optical satellite data processing
Optical image sub-pixel correlation of aerial photographs and satellite images has been extensively employed to quantify ground deformation associated with coseismic fault ruptures. This method is particularly effective for measuring surface displacement in near-field regions, where InSAR techniques often encounter decorrelation due to high strain gradients[56–58,67–69]. In this study, we applied OIC to investigate the Mandalay earthquake on 28 March 2025, using Copernicus Sentinel-2 Level-2A imagery at 10 m spatial resolution with minimal cloud cover. Specifically, we correlated post-earthquake Sentinel-2 images acquired on 30 March and 1 April 2025 with pre-earthquake images taken on 2 and 20 March 2025. The short temporal separation (~1 month) between pre- and post-event images minimized land-cover changes, thereby ensuring high-quality image correlation. However, given that the post-earthquake images were acquired several days after the event, our displacement measurements reflect both coseismic deformation and a limited component of early post-seismic deformation. We employed the frequency-domain correlator implemented in the COSI-Corr software[58], which computes correlation in the Fourier domain rather than the spatial domain. This approach offers three primary advantages: (1) improved robustness against radiometric differences between images due to illumination and atmospheric variations, (2) sharper correlation peaks resulting in enhanced sub-pixel accuracy (~0.05 pixel), and (3) higher computational efficiency suitable for processing large satellite imagery datasets. We used a correlation window size of 64 pixels and a correlation step of 4 pixels. This procedure generated continuous 2D displacement maps (east-west and north-south) with an effective spatial resolution of approximately 40 m[56–60]. A linear ramp derived from non-deforming areas far from the fault was removed from these displacement fields, and unrealistic displacement values exceeding ±5 m were discarded. Although applying additional smoothing filters (e.g., median or Non-Local Means filters) could reduce noise further, we chose to analyze the raw correlation data to preserve detailed small-scale deformation features

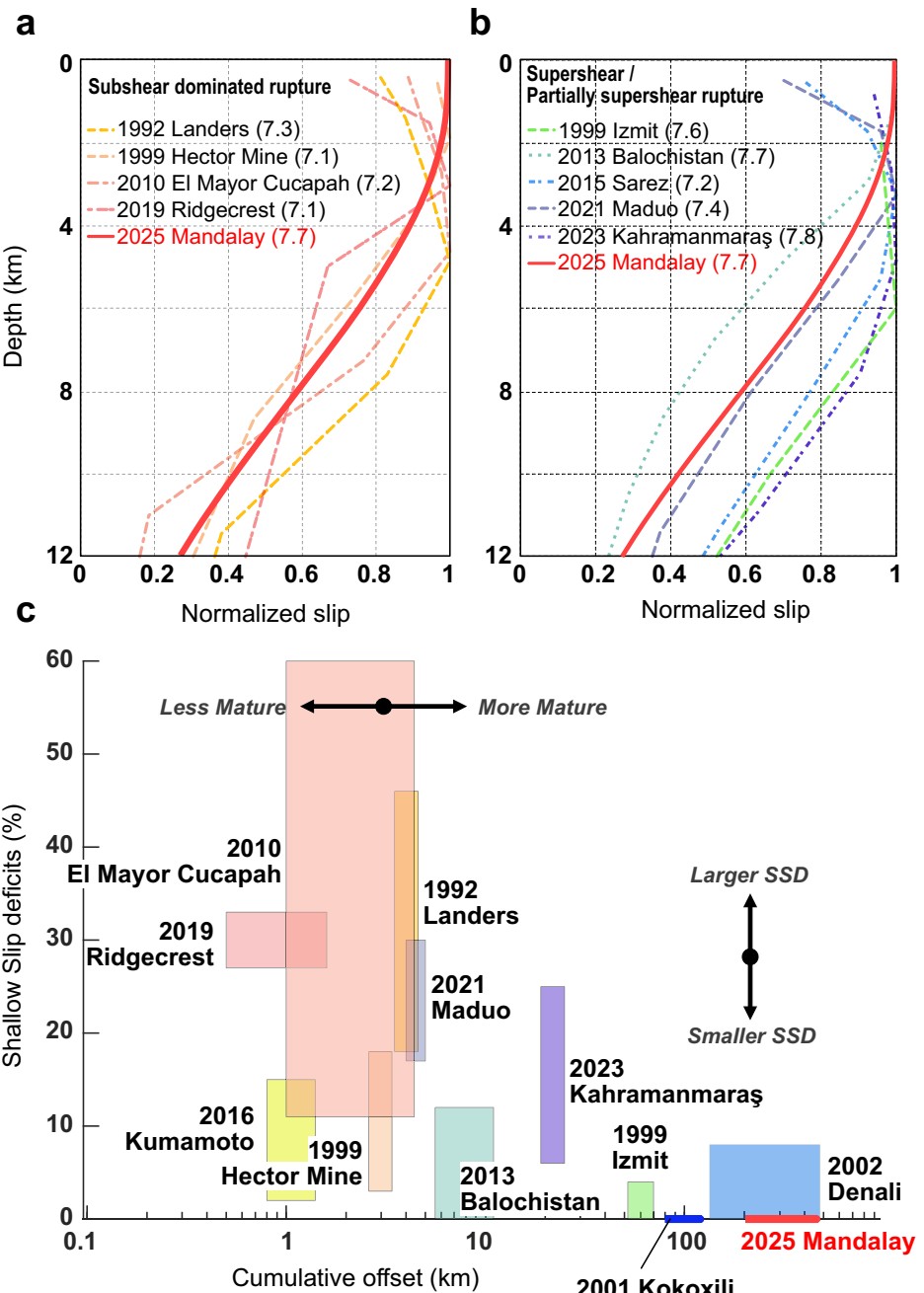

**Fig. 5 | Comparison of SSD inferred in previous large strike-slip earthquakes and the 2025 Mandalay earthquake. a** compares the average slip profile vs. depth from our preferred model (Fig. 4) to profiles from recent sub-shear events (References in supplementary Table S1); **b** compares this event to recent supershear or partially-supershear events. **c** shows the correlation between cumulative fault offset and total shallow slip deficit. The 2025 Mandalay earthquake is at the extreme end of this scale, with a cumulative offset similar to the Denali fault (200–500 km) and no detectable shallow slip deficit. A future rupture along geometrically simple segments of other mature faults would likely fall near this line, while ruptures along geometrically complex fault segments are likely to exhibit significant off-fault deformation (OFD) and shallow-slip deficit (SSD).

adjacent to the fault trace; the final North-South and East-West displacement fields are shown in Supplementary Fig. S1.

To characterize fault displacement systematically, we extracted 177 displacement profiles perpendicular to the previously mapped fault trace. Each profile was defined as a 16 km-long and 2 km-wide swath centered on the mapped fault line. Within each profile, we computed linear trends independently for areas located 0.8–8 km on each side of the fault, using least-squares fitting. The displacement offset across the fault was then determined by calculating the difference between these two linear trend values at the position of the mapped fault trace. These profiles provided a dense and consistent dataset for analyzing detailed spatial variability in surface slip along the fault (Fig. 3).

### SAR data processing

To complement the near-field measurements from optical image correlation and provide broader spatial coverage of the coseismic deformation, we processed Interferometric Synthetic Aperture Radar (InSAR) data. InSAR is highly sensitive to ground displacement in the satellite's line-of-sight (LOS) direction and is effective for mapping deformation patterns over large areas [Simons & Rosen, 2015]. We used C-band Interferometric Wide (IW) swath data acquired by the

Sentinel-1A satellite[70]. Four twelve-day interferometric pairs spanning the earthquake were processed, from descending track 33 (pre-event: 2025/03/19, post-event: 2025/03/31) and track 106 (pre-event: 2025/03/24, post-event: 2025/04/05), and ascending track 70 (pre-event: 2025/03/22, post-event: 2025/04/03) and track 143 (pre-event: 2025/03/27, post-event: 2025/04/08). The wrapped interferograms are shown in Supplementary Fig. S2.

Interferometric processing was carried out using the open-source ISCE2 software package[71], following the standard workflow optimized for Sentinel-1 TOPS data[72]. This included precise coregistration of Single Look Complex (SLC) images based on Enhanced Spectral Diversity[73], generation of differential interferograms using the 1 arc-second (~30 m) SRTM digital elevation model to remove topographic phase[74–76], and multilooking (nine looks in range, three in azimuth) to enhance the signal-to-noise ratio. Phase unwrapping was performed using the Statistical-cost Network-flow Algorithm for Phase Unwrapping (SNAPHU)[77]. Recognizing that high strain gradients and surface disruption near the fault trace can introduce unwrapping errors, only the largest connected component of the unwrapped phase field was retained[78], and we additionally masked pixels within a ~1 km buffer surrounding the fault. The resulting unwrapped interferograms from the descending tracks, representing coseismic displacement in the satellite LOS (positive = range increase), are shown in Supplementary Fig. S3.

In addition to interferometric phase analysis, we computed pixel offsets in both the range (satellite LOS) and azimuth (satellite flight direction) directions using the amplitude cross-correlation capabilities within ISCE[79]. This technique is less precise than InSAR phase analysis but is robust in areas of high deformation gradients where InSAR coherence is lost, making it highly complementary for capturing near-fault displacements[80,81]. We used standard correlation windows of 64 by 64 pixels with an oversampling factor of 32 to achieve sub-pixel accuracy; values with a signal-to-noise ratio less than 30 were discarded. Both the unwrapped interferograms and the offset maps were geocoded at 30 m resolution. Similar to the optical image correlation data processing, long-wavelength signals potentially related to residual orbital errors or atmospheric artifacts were estimated and removed by fitting and subtracting a bilinear ramp from areas far from the fault zone. The final azimuth and range offset maps are shown in Fig. 2 and Supplementary Figs. S4 and S5.

### Finite-fault inversion

To infer the distribution of slip on the fault plane, we performed a coseismic slip inversion using the geodetic datasets. We represented the fault geometry as a single continuous surface extending approximately 700 km along the mapped surface trace of the Sagaing fault, from 17°N to 23.25°N. The fault plane was discretized using a triangular mesh comprising 4216 patches with variable size, increasing from an average linear dimension of 1–2 km near the surface to 4–6 km at a depth of 20 km. We employed Green's functions for triangular dislocations in a homogeneous elastic half-space[82]. To regularize the inversion and ensure a physically plausible slip distribution, we applied smoothing based on the shear stress kernel, which approximates the Laplacian operator but permits interaction between all patches and more accurately accounts for both patch size and the proximity of the free surface[83]. The optimal magnitude of the regularization strength was determined using an L-curve (Supplementary Fig. S6); a comparison of models with higher and lower smoothing is shown in Supplementary Fig. S10. The primary effect of varying smoothing is to change the apparent depth extent of the rupture, due to relatively poor data constraints in this part of the fault. Models with lower smoothing have even shallower average slip than our best-fitting model, indicating the true fault slip could be extremely shallow.

We identified the best-fitting fault dip along the northern and southern parts of the rupture by conducting a grid search over two parameters for the dip angle in the northern and southern parts of the fault, with a linear variation in dip assumed across a 200 km length centered at 20°N. The best-fitting model has a dip in the northern segment of 65°E, and 80°W in the southern segment (Fig. S7), in surprisingly good agreement with previous interseismic geodetic inferences in the northern part[22,23]. The slip distribution for both a vertical and dipping model is highly similar (Fig. S8) but the data fit is significantly better for a dipping fault. The preferred fault geometry is shown in map view in Fig. S9. Our final model allows only right-lateral strike-slip; we also constructed a model including dip-slip (Fig. S11) but found that it did not significantly change the slip distribution or improve the data fit. Furthermore, the increased number of degrees of freedom in the 2-component model results in a higher reduced-chi-squared value that can be rejected by an F-test with likelihood ~ $10^{-275}$ (see Fig. S11 caption). We computed the seismic moment, magnitude and average stress drop for our models assuming a constant shear modulus of 30 GPa, but we note this could be an overestimate given the shallow nature of slip in this event. A value of 25 GPa would reduce the magnitude for our best fitting model from 7.75 to 7.7, in closer agreement with seismic observations[40].

Input data for the inversion consisted of the north-south component of the Sentinel-2 optical image correlation, azimuth and range offsets from all four Sentinel-1 tracks, and the unwrapped, masked LOS displacement from the two descending Sentinel-1 interferograms (tracks 33 and 106). The datasets were downsampled using a quadtree algorithm[80,84], varying the sampling density based on distance from the fault trace to prioritize near-field information while reducing computational load. Optical and SAR offset datasets were limited to 50 km distance from the fault, owing to the inherent data noise exceeding the signal beyond this distance and introducing artifacts. InSAR data were included up to 100 km distance owing to their lower intrinsic noise, to provide better constraints on the deeper part of the fault. This resulted in a total of 15,130 observation points to be used for the inversion across the eleven datasets. Data points were weighted inversely by the standard deviation of the original pixels within each downsampling quadtree cell, resulting in an average uncertainty of 36 cm for the optical data, 35–48 cm for the azimuth offsets, and 6–9 cm for the range offsets, owing to Sentinel-1's higher range resolution. The InSAR phase uncertainties derived from this method were unreasonably small, less than 1 cm, and a few pixels within the other datasets had similarly unreasonably small values. Therefore, we imposed a minimum uncertainty floor of 3 cm for all datasets (10 cm for the track 106 phase, which was strongly affected by localized tropospheric disturbance near Mandalay) to prevent over-fitting of the InSAR and individual data points. The downsampled datasets, model fits, and residuals are shown in Supplementary Figs. S13–S23. The ascending InSAR interferograms were excluded from the preferred model; while broadly consistent with the other datasets, they are strongly affected by tropospheric noise, likely associated with thunderstorms which are common in Myanmar at ~ 6 pm, the local image collection time of for ascending Sentinel-1 orbits.

To account for potential residual long-wavelength signals (e.g., orbital errors, atmospheric effects) in the geodetic data, we simultaneously estimated three additional parameters defining a bilinear ramp for each dataset alongside the slip on the fault patches. Such ramps can trade off with slip on the deeper fault patches, so we added an additional small penalty applied to the total magnitude of slip on each patch[83,85]. The weight applied to this penalty was set to be one order of magnitude smaller than the smoothing penalty, discouraging deep slip from solely fitting long-wavelength trends while allowing geologically reasonable slip patterns.

### Data availability

The Sentinel-1 SAR imagery and Sentinel-2 optical imagery are Copernicus Sentinel data products [2025] and the raw data are freely

available (https://dataspace.copernicus.eu/). Our final, processed pixel offset maps from Sentinel-2 and Sentinel-1 Azimuth/Range offsets and wrapped/unwrapped phase are available at https://doi.org/10.5281/zenodo.17393910.

## Code availability

We processed the data using the open source software packages COSI-Corr (https://www.tectonics.caltech.edu/slip_history/spot_coseis/download_software.html) and ISCE version 2.6.3 (https://github.com/isce-framework/isce2). The modeling code, including all input files and preprocessing steps for this event, is included at https://doi.org/10.5281/zenodo.17393910, and at https://github.com/ericlindsey/stress-shadows/.

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

## Acknowledgements

We thank Ei Mhone Nathar Myo and Saw Myatmin for helping us to collect the information on the earthquake and damage in the first moment. We thank Tai-Lin Tseng for valuable scientific discussion. This study was supported by the National Science and Technology Council of Taiwan under grant numbers 113-2116-M-194 -008 to Y.T., 112-2116-M-002 -007 and 113-2116-M-002 -028 to Y.W.

## Author contributions

E.O.L., Y.W., and Y.-T.K. conducted the study and wrote the manuscript. Y.-T.K. and Y.W. processed the Sentinel-2 data. E.O.L. processed the Sentinel-1 data and performed the fault slip inversions. E.O.L. and Y.W. performed the SSD analysis. All authors contributed to the discussion and interpretation of the results.

## Competing interests

The authors declare no competing interests.
