## [Transparent Peer Review file · Nature Communications]

Mature fault mechanics revealed by the highly efficient 2025 Mandalay earthquake

Corresponding Author: Dr Yu Wang

Version 0:

Reviewer comments:

Reviewer #1

(Remarks to the Author)

In this study the authors use InSAR and optical correlation data to infer a static slip model for the 2025 Mw 7.7 Myanmar earthquake. They found an exceptionally long and geometrically simple fault and a relatively smooth slip distribution with no Shallow Slip Deficit (SSD). They interpret the absence of SSD as a result of the high maturity level of the Sagaing fault. The slip model reveals lower slip amplitudes on segments that previously ruptured in 1929, 1930, and 1956, which is consistent with a slip-predictable behavior. Furthermore, they find that the surface slip amplitude exceeds the theoretical slip deficit expected to have accumulated since the last major earthquake in 1839. They propose this could be attributed to a dynamic overshoot or, alternatively, may suggest that the 2025 earthquake released stress accumulated over multiple seismic cycles — stress that the 1839 event may not have fully released. The paper is technically sound, well written, and easy to follow. The figures are clear, although the captions for some of them are incomplete. I have the following comments:

1. Inverse correlation between SSD and fault maturity:

I am a bit skeptical of the claim that there is a strong inverse correlation between shallow slip deficit (SSD) and fault maturity. This relationship has been discussed in several previous studies (e.g., Dolan & Haravitch, 2014; Sethanant et al., BSSA 2023; Antoine et al., 2024), and recent work suggests that earthquake magnitude is the primary controlling factor. For example, Sethanant et al. (2023) compare SSD with both earthquake magnitude and fault maturity for 28 events and find a strong correlation between SSD and magnitude, but only a very weak correlation with fault maturity. This is also clearly shown in Figure 3 of Antoine et al. (2024). The apparent correlation between SSD and fault maturity may in fact reflect the tendency of highly mature faults to be geometrically simple and therefore more likely to produce very large earthquakes. In contrast, I am not aware of any moderate earthquake (e.g. Mw 6.7-7.2) that occurred on a mature fault and did not show any SSD. Overall many factors influence the amount of shallow slip, and attributing it solely to fault maturity is an oversimplification.

2. Depth extent of slip

Looking at the supplementary figures, it seems that most of the datasets does not sample the decrease of displacement away from the fault (there is not many far-field data points). I am therefore wondering if the depth extent of slip is well constrained by the data?

2. Fault geometry:

The authors argue that no significant asymmetry is observed in the surface displacement field and therefore adopt a vertical fault geometry. However, looking at the azimuth and range offsets, it seems to me that the surface displacement field is slightly asymmetric and suggests an eastward dipping fault. Additionally, the aftershocks are predominantly located on the eastern side of the fault trace, which would also support this interpretation. I suggest that the authors consider inverting for fault geometry, as assuming an incorrect fault dip could bias the inferred slip distribution at depth.

3. Aftershocks distribution

L 96-98 and Figure 1. The statement that aftershocks are primarily located at the edges of the rupture should be somewhat

tempered. As shown in Figure 1, the southern cluster of aftershocks extends along more than one-third of the rupture length, while the northern cluster appears is not strongly more concentrated than the distribution observed in the central section. Furthermore, it is unclear whether the apparent lack of aftershocks in the central part of the rupture is real or could be due to limitations in detection capability. Could the authors add the station locations on Figure 1? Additionally, it is notable that all aftershocks are located on the eastern side of the fault. This spatial pattern deserves further discussion. Does it reflect fault geometry or other structural or detection-related factors?

Minor comments:

L 96-98 "indicative of a smooth rupture and relatively uniform stress drop." Please add references

L.111-112: I do not see any clear bend in the southern part of the rupture on Figure 2. It might be more appropriate to refer to Figure 1, where the bend near 18°N is indeed visible.

L 335: datasets  datasets

Figures:

Figure 1: It is difficult to distinguish the aftershocks from the background seismicity. I suggest using a distinctly different color for the aftershocks to improve visual clarity. Since the aftershocks all appear to be shallow (i.e., <10 km), it may not be necessary to color them by depth. Using a single color clearly different from those used for background events would make them stand out more effectively.

Figure 2: For consistency, please also indicate the dates of the pre- and post-event Sentinel-2C images, as is done for the other datasets.

Figure 3: While the meaning of the yellow star and red dashed lines can be easily inferred, the black solid and dashed lines are not explained and should be clarified. Additionally, the abbreviation "MDL" is not defined.

Figure 4: Caption for panel B is missing.

Supplementary data:

I am not sure to understand Figure S1. It is described as showing the raw optical correlation results, but the data coverage seems to be different from that shown in Figure 1?

(Remarks on code availability)

Reviewer #2

(Remarks to the Author)

This contribution is a timely analysis of the 2025 Sagaing earthquake focusing on InSAR and Sub-pixel optical image matching. As far as I can see no journal articles applying these techniques to this event have yet been published. The authors highlight the lack of a shallow-slip deficit and comment on the magnitude of slip in terms of the geodetic slip-accumulation rate and different models of fault slip-accumulation and release. The techniques used are appropriate and I believe the results are interesting and valuable. The interpretation is suitably cautious and potential inconsistencies are pointed out. The main weakness of the manuscript is the lack of constraints from seismology. I believe that there is an emerging consensus that this is a super-shear event, a fact which could have implications for their interpretation. I do not think that the lack of an original seismological analysis should prevent publication, but some comment on the possible role (or lack of role) of supershear rupture is probably necessary. The authors clearly considered this as there is some consideration in a figure, but this does not seem to have made it through to the text. Some more comment on potential segment boundaries may also be useful and the authors may consider providing a little more analysis on the mismatch in dip between their model and the seismological results.

The text is very clear and well-written, but I have a few minor comments

Specific comments

72-74: I would normally expect 'segments' to mean distinct sections of a fault separated by some structural complication (e.g. a bend or step-over) which may therefore be expected to rupture separately over multiple earthquake cycles. Is that the case here? The detailed fault morphology (in this sense) is not presented very clearly. It wouldn't be surprising for a straight section of fault (a single segment) to occasionally fail in small events before a larger event ruptures its entire length. The Robinson et al. paper cited for the simplicity of the fault is also not very detailed. Is there a more focused study on the fault morphology?

83-84: What defines the ends of this section?

84-85: The cited paper is a bit more cautious in its assignment of the causative fault of the Ava earthquake than is implied here, but I agree that the 2025 event does make it likely that this interpretation is correct.

94-96: The USGS finite fault model (version 3) states that the earthquake was supershear.

<https://earthquake.usgs.gov/earthquakes/eventpage/us7000pn9s/finite-fault>

Recent research publications confirm this (e.g. Inoue, N., Yamaguchi, R., Yagi, Y., Okuwaki, R., Bogdan, E., & Tadapansawut, T. (2025). A multiple asymmetric bilateral rupture sequence derived from the peculiar tele-seismic P-waves of the 2025 Mandalay, Myanmar earthquake. *Seismica*, 4(1). <https://doi.org/10.26443/seismica.v4i1.1691>)

There does not appear to be much discussion of the potential role of supershear rupture other than the separation of profiles in Fig 5a/b. I think there needs to be some treatment of this in the text.

119-120: Also state the length and limiting latitude to North.

130-133: Was the effect of the fault dip assessed quantitatively or was the dip fixed at 90° due to lack of pronounced asymmetry. I would not expect a steeply dipping N-S pure strike-slip fault to be strongly asymmetrical (e.g. Sethanant, I., & Nissen, E. (2025). The InSAR lookbook: an illustrated guide to earthquake deformation interferograms. *Seismica*, 4(1). <https://doi.org/10.26443/seismica.v4i1.1413>) and given the fact that the seismological estimates do suggest a non-vertical dip it seems important to at least quantitatively test the geometry they prefer.

156-159: This could be possibly be rewritten more clearly.

187: Could you include a statement here about the horizontal resolution of your method (i.e. how distributed would OFD need to be to show up as such). This is in the methods, but I think it needs to be briefly stated here as well.

195: Also a supershear event?

201-202 Would you be able to resolve OFD at the lower end of this scale.

208-209: I have some concerns about this statement because it is not that certain exactly what failed in this event, but it does seem likely the area that has the highest slip did so I suppose this comment is reasonable.

347-350: Doesn't this belong in the results rather than the methods?

(Remarks on code availability)

Reviewer #3

(Remarks to the Author)

The authors present an analysis of displacement data from the 2025 Mandalay, Myanmar earthquake with a goal of modeling the event and drawing inferences about fault behavior and properties. Using surface displacement estimates from optical image correlation, radar image correlation and InSAR, they produce a dislocation model of the slip in the earthquake, and highlight several atypical features of that model compared with other events from the past few decades: 1) along much of the fault, peak slip is at or close to the surface; 2) the fault rupture was unusually long for an earthquake of this size; 3) slip is quasi-slip predictable, in that areas of the with the longest time since the previous earthquake rupture slipped more than areas that had more recent ruptures; 4) the slip in some areas was greater than expected, given our estimates of fault slip rate and time since the last event.

The first of these findings leads the authors to make a broader point about the shallow slip deficit (SSD) for this event – or apparent lack of it. Compared with other events of a similar size and scale for which similar geodetic data are available, the Mandalay earthquake had significantly larger shallow slip, and a negligible SSD. In previous studies of SSDs of earthquakes, events that have large SSDs typically have surface ruptures that are distributed over wide fault zones (often hundreds of meters wide), with significant off-fault deformation accommodated by inelastic processes. In contrast, the authors suggest, the minimal SSD of the Mandalay event implies that the fault zone is highly localized and likely a result of a very mature fault, with minimal roughness and segmentation. Another interesting inference drawn from the model results include the possibility of some kind of 'dynamic overshoot' – a transient stress effect of the earthquake rupture – being a potential explanation for the slip in the earthquake exceeding the prediction based on slip rate and interval since the last earthquake.

Overall, this is a solid study that provides an interesting end-member data point in the literature about SSDs. The conclusions made are reasonable, and entirely consistent with the results presented. The text and figures are clear. I have a few questions about choices that were made in the modeling, but I do not think they will substantively change the findings of the study (but they might change some of the details), and I do not think they would require major corrections to address. I will start with my longest comments, and then move to line-by-line comments.

1) Assumptions made in the dislocation model

One thing that I noted is that the authors' model has some baked-in assumptions that ought to be evaluated – that the fault dip is vertical, and that the rake is pure right-lateral. This differs from the USGS W-phase and Global CMT solutions, which could be considered 'average' solutions of the whole event, both of which suggest the fault dips to the east and has a minor dip-slip component of motion. One consequence of using this assumed geometry is that it could result in biases to the slip distribution – more slip could be required to match the observed surface deformation, for instance. This could be an explanation for the estimated moment magnitude from the authors' model exceeding seismic estimates by ~0.1 magnitude units (I cannot say exactly what the difference is in seismic moment, as I could not find that number in the manuscript – I would urge that such information be included). Indeed, I do not recall the authors commenting upon that difference.

It would be helpful for the authors to assess a few other fault geometries (e.g. the W-phase solution) to see what effect they have on the modeled slip, and also how well such models fit the data. I do not expect this would change the primary conclusion surrounding the SSD, as the estimated slip values from an alternate fault geometry I are likely to vary in proportion to the values in the authors' current preferred model – the portions of the fault that slip the most will still be the

portions that slip the most, shallow parts of the fault will still slip more than deeper parts, the SSD will still be negligible – but some of the secondary conclusions about the possible 'overshoot' of slip in the earthquake may be affected if the estimated slip changes (if it goes down, it could mean there is no overshoot... but if it goes up, then there could be more of an overshoot).

2) How sensitive is the model to shallow slip?

If I read correctly, it seems that the fault geometry has a minimum element size of 1–2 km. I understand that this is a long rupture, and there are practical considerations about how small the elements can be for the inverse problem to be tractable. But one of the things I wonder is what the limits of the model could be for resolving shallow slip, and any deficits therein? What scale of SSD would be impossible to detect with the preferred fault geometry? I still think the shallow slip pattern is highly suggestive of the authors' conclusions, but it would be good to place limits on what can be said in that regard – perhaps with some kind of synthetic test?

3) Line-by-line comments

Line 64: Is Istanbul an appropriate comparison? How mature are the faults in the Sea of Marmara? My thought is that they are probably less mature (and straight) than the San Andreas. I also wonder where else in the world there are appropriately mature strike-slip faults near major cities? Guatemala City? Wellington, New Zealand? Any thoughts?

Line 85: Approximately what was the "full slip" of the 1839 earthquake? It would be very interesting to know how it compares with the slip in this event, if that is at all possible?

Line 129: It may be that changes in dip cannot be resolved, but I do not think that a consistent, non-vertical dip is ruled out by a lack of asymmetry.

Lines 208–211: It occurs to me that if dynamic overshoot is a consistent feature of earthquakes on this portion of the Sagaing fault, then it should be included in geologic slip rates, but perhaps not in geodetic estimates of slip rate. It seems that all of the slip rate estimates quoted here are geodetic? If this reasoning is correct, then the difference between estimated and actual slip is in the direction you would expect, I think?

Lines 320–323: A single fixed rake that was not pure right-lateral would have the same number of degrees of freedom as the authors' preferred model though, so if it fit the data better, it would be significant. So it isn't "dip-slip" that has been ruled out, so much as "variable dip-slip".

Lines 332–333: My hunch is that 3 cm is pretty small as minimum errors in image matching go (it is low for InSAR too, but possibly defensible)? How did the authors arrive at that number?

Line 349: Please quote the actual seismic moment estimate for the model, either here or somewhere else in the manuscript.

Figure 3b: Why not plot local fault-parallel offset here, rather than east and north separately? "conponent" in the plot legend is a misspelling

Figure 3c: Same comment as above about what is plotted here. Also, these profiles should be primary evidence for a lack of SSD. You would want to confirm that the peak offset is at the fault – is it? In Profile 21 in particular it looks like there could be some off-fault deformation, on the order of a few 100 m? (That would be in keeping with other studies where it has been identified, I think?) It is hard to tell exactly as the profiles are so zoomed out (each division on the scale is 2 km, usually SSDs are identified on a much shorter length scale than that).

(Remarks on code availability)

I think the results would be potentially reproducible if the appropriate input files were provided, but there seems to be no link to the Zenodo repository in the manuscript document (although one is mentioned). The GitHub repository linked to does have test cases for the code.

Version 1:

Reviewer comments:

Reviewer #1

(Remarks to the Author)

The authors have addressed all of my questions thoroughly and clearly. I believe the manuscript is now in excellent shape, and I am pleased to recommend it for publication in its current form.

Reviewer #2

(Remarks to the Author)

The authors have addressed my earlier comments fully and I believe that manuscript has been significantly improved

(especially by investigating the fault dip as requested by all reviewers). I believe that the manuscript is ready for publication.

Reviewer #4

(Remarks to the Author)

This is a very interesting paper that explores the slip pattern in the 2025 Mandalay Earthquake. The authors use geodetic data to invert for the slip in this event, and find that it appears to have experienced peak slip right up at the free surface, with very little off-fault deformation (OFD). This result contrasts with many recent earthquakes, which display a shallow slip deficit and significant OFD. The authors attribute this result to the relatively straight, mature nature of this fault, and show that other recent earthquakes on faults with similar structures have displayed similar slip patterns. The results could have important implications for the near-fault deformation and damage, as well as radiated ground motion.

I found the paper straightforward to read, and their conclusions seem plausible. I don't have any major criticisms; my comments are largely editorial in nature. I think the paper will be publishable with minor revisions. I list below my line-by-line comments.

22: "Focused" is a rather vague term; could the authors use something more specific, or define how they are using this term?

29, 72-73: Please clarify what is meant by "efficient" in this context. Many readers will assume it means radiation efficiency, but not having read the rest of the paper yet, it's not clear.

129: I believe panel (d) is mislabeled as (e) in the caption to figure 2.

204-205: This appears to be the definition of "efficiency" in this paper; it would be very good to provide it much earlier in the paper, the first time the term is used.

218: Do you have a correlation coefficient or other numerical factor to quantify this inverse correlation? I can see it by eye in the diagram, but if it's just a qualitative correlation, that should be made explicit.

243: Hu et al. don't question the presence of dynamic overshoot in strike-slip settings. There is no obvious reason why it wouldn't occur there, as it is a physical manifestation of the momentum of the fault as it slides. In models it is typically a relatively small fraction of the entire slip, though.

254: I'd be wary about drawing too many conclusions from a single event. Figure 5 shows that other earthquakes, with similarly simple fault traces, may also have these properties. The conclusions would be strengthened if they mentioned these other corroborating events.

We thank you for providing three timely, detailed and positive reviews to our manuscript. We have carefully revised the manuscript in response to all reviewer comments. In addition to the text changes described below and highlighted in the track-changes manuscript, we have added two new supplementary figures (Figures S7, S8 and S9) and updated all other model-related figures after changing our preferred fault geometry to a dipping one, as suggested by the reviewers. Our replies to those of each reviewer are listed in blue text below; line numbers refer to the changes-tracked version of the manuscript.

In particular, we have replaced the originally vertical fault model with a dipping one, after conducting a parameter search for the optimal dipping values in the northern and southern parts of the fault, which we find to be in close accord with existing literature. The revised model (described below and in the revised text) now fits all our datasets significantly better as well. However, the pattern of slip is largely unchanged and does not affect our primary conclusions regarding the shallow slip deficit. We feel this test significantly strengthens our results and we thank the reviewers for suggesting it.

We have also carefully revised the text to more carefully address the issue of fault maturity and resolution of the model in the shallow subsurface, in addition to the other concerns noted below. We hope you and the reviewers will find these changes acceptable.

REVIEWER COMMENTS

Reviewer #1 (Remarks to the Author):

In this study the authors use InSAR and optical correlation data to infer a static slip model for the 2025 Mw 7.7 Myanmar earthquake. They found an exceptionally long and geometrically simple fault and a relatively smooth slip distribution with no Shallow Slip Deficit (SSD). They interpret the absence of SSD as a result of the high maturity level of the Sagaing fault. The slip model reveals lower slip amplitudes on segments that previously ruptured in 1929, 1930, and 1956, which is consistent with a slip-predictable behavior. Furthermore, they find that the surface slip amplitude exceeds the theoretical slip deficit expected to have accumulated since the last major earthquake in 1839. They propose this could be attributed to a dynamic overshoot or, alternatively, may suggest that the 2025 earthquake released stress accumulated over multiple seismic cycles—stress that the 1839 event may not have fully released. The paper is technically sound, well written, and easy to follow. The figures are clear, although the captions for some of them are incomplete. I have the following comments:

Thank you for the detailed and overall positive comments!

1. Inverse correlation between SSD and fault maturity:

I am a bit skeptical of the claim that there is a strong inverse correlation between shallow slip deficit (SSD) and fault maturity. This relationship has been discussed in several previous studies (e.g., Dolan & Haravitch, 2014; Sethanant et al., BSSA 2023; Antoine et al., 2024), and recent work suggests that earthquake magnitude is the primary controlling factor. For example, Sethanant et al. (2023) compare SSD with both earthquake magnitude and fault maturity for 28 events and find a strong correlation between SSD and magnitude, but only a very weak correlation with fault maturity. This is also clearly shown in Figure 3 of Antoine et al. (2024). The apparent correlation between SSD and fault maturity may in fact reflect the tendency of highly mature faults to be geometrically simple and therefore more likely to produce very large earthquakes. In contrast, I am not aware of any moderate earthquake (e.g. Mw 6.7-7.2) that occurred on a mature fault and did not show any SSD. Overall many factors influence the amount of shallow slip, and attributing it solely to fault maturity is an oversimplification.

Thanks for the critical and helpful comments. We agree that the full cause of shallow slip deficit in a given case is likely complex; fault maturity, geometric complexity, and earthquake size are inter-related, making this issue tricky to disentangle. Although all these factors are clearly important, our claim is that fault maturity provides a unifying perspective that clarifies the causal relationship – increasing maturity leads to smoother faults, which rupture in larger earthquakes and have less shallow slip deficit. We have revised the text throughout the final section (titled “Near-total absence of shallow slip deficit”, **line 271**) to make this perspective more clear with

respect to two points: (1) the SSD vs. fault maturity relationship, and (2) the earthquake size vs. fault maturity relationship. Some additional discussion and notes regarding what we changed follows:

Regarding (1), the key difference between our analysis and that of Sethanant et al. (2023) is that we only consider earthquakes with larger magnitudes ($M > 7$) and exclude those with smaller magnitudes. As pointed out by both Sethanant et al. (2023) and Antoine et al. (2024), earthquakes with larger magnitudes are those most likely to rupture the entire seismogenic width of the fault, and generally have smaller SSD overall; so by limiting our analysis to these events we are better able to isolate the fault maturity signal from this 'saturation' effect. If we exclude events with $M < 7$ from the data plotted by Sethanant et al. (their Table 3) we find that their data shows SSD decreases when total fault offset increases, as we found in our Figure 5. Also, the trend between SSD-magnitude and SSD-log(maximum total offset) is very similar. Our values (Table S1) are slightly different than those from Sethanant et al., because we use a different set of updated SSD estimates from recent publications, including the values compiled by Antoine et al. (2024), but the relationship is similar.

[editorial note: third party material redacted]

To more clearly reflect this nuance with regard to earthquake size, we added the following sentence at **line 300**: “Although previous studies suggest a wide range of SSD values and a complex cause of the SSD, particularly in smaller ($M < 7$ events)¹⁵, we find a strong inverse correlation between SSD and fault maturity in large ($M > 7$) events, with the Mandalay earthquake at the extreme end of the spectrum.”

It is true that there are some earthquakes with a large SSD that occurred on highly mature faults; one notable event from Sethanant et al.’s analysis is the M6.7 2002 Nenana Mtn earthquake (Wright et al., 2003), which occurred on a fault with ~300km offset and yet had an 86% SSD. Our analysis holds only for larger events on these faults and does not preclude smaller events like this. Our statement therefore now includes the requirement “large ($M > 7$) events” to exclude smaller events like this. (**line 318**)

With regard to (2), when we plot the data from Sethanant et al. (2023)’s Table 3 in another way, there is a good correlation between earthquake magnitude and total fault offset on a log scale, our proxy for fault maturity (figure below). This suggests that the earthquake magnitude could be affected by fault maturity. This makes sense, in that mature faults tend to be longer, smoother, simpler, and more continuous, creating an environment in favor of larger earthquakes, though of course events like the Nenana Mtn earthquake are a common exception. Hence, unlike the conclusion drawn by Antoine et al. (2024) that fault maturity only plays a secondary role in controlling the shallow slip distribution and the occurrence of SSD, we argue that the fault maturity is the root cause of both larger earthquakes and lower SSD.

[editorial note: third party material redacted]

Data from Sethanant et al. (2023)’s compilation in their Table 3, showing the relationship between earthquake magnitude (Y-axis) and maximum estimated total fault offset (X-axis). This plot shows a good correlation between the earthquake magnitude and the fault maturity.

Our overall perspective that maturity is the underlying cause of both low SSD and larger magnitude events is best captured in the modified sentence on **line 320**: “highly mature faults tend to develop a simple and smooth fault trace, favoring the occurrence of large earthquakes

with minimal SSD”.

2. Depth extent of slip

Looking at the supplementary figures, it seems that most of the datasets does not sample the decrease of displacement away from the fault (there is not many far-field data points). I am therefore wondering if the depth extent of slip is well constrained by the data?

Thanks for this comment. Partly this could be an issue of the extreme aspect ratio of this rupture, making the data look closer to the fault than it really is – most of the datasets extend ~50 km from the fault along their full length, which is very wide for imagery datasets in this setting.

Still, there are some limitations to our model’s resolution of the depth of slip, and we have tried to better quantify them. In particular, Supplementary Figure S10 compares our preferred model to two others with greater and lesser smoothing strength, new version copied here:

Updated Supplementary Figure S10, showing varying depth extent of slip based on the assumed smoothing.

Models with lower smoothing (which should better fit the data) generally have shallower overall slip, indicating that possibly the true depth extent of slip is even more shallow than in our preferred model, but the true depth extent is difficult to resolve accurately given this tradeoff.

There is some difficulty in obtaining data constraints farther from the fault in some areas, particularly in the south, where dense forest approaches up to the fault trace on the western side. As a result, none of our datasets fully constrain the displacement on the western side of the fault between 18°-19°N. However, in the remainder of the fault our datasets have strong correlation up to ~50km from the fault, which is approximately ~5 times the depth of slip, and the

interferometric phase (supplementary figures S22 and S23), which have the lowest uncertainties, extend up to 100 km from the fault. We limited the data to these distances due to the greater contribution of long-wavelength errors in the datasets (orbital, ionospheric, and tropospheric) at longer distances, particularly in the range and azimuth offsets, which we found introduced non-physical biases in the deep slip.

One additional note regarding the depth extent of slip – the dipping model generally fits the far-field data much better with less deep slip than required in the vertical model, as can be seen in the comparison with the vertical model (new Figure S8), reproduced below:

New Supplementary Figure S8, comparing our new dipping model (a) to the old vertical model (b). Of note here is the reduced slip at depth in the northern part (22.5 – 23.3°N). This deep slip in the vertical model was required to fit the asymmetry in the far-field data, which is now adequately explained (with a better overall fit) by the fault dip.

To better address these considerations and make the text more consistent with our updated model, we have updated the text as follows:

Line 214: changed “with the depth extent gradually decreasing **southward**” to “with the depth extent gradually decreasing **at the southern end**”.

Line 234: deleted “To the south of the epicenter, the rupture is generally shallower than to the north, with the majority of slip between the surface and 10 km depth.”

Line 512: added “The primary effect of varying smoothing is to change the apparent depth extent of the rupture, due to relatively poor data constraints in this part of the fault. Models with

lower smoothing have even shallower average slip than our best-fitting model, indicating the true fault slip could be extremely shallow.”

Line 549: added “Optical and SAR offset datasets were limited to 50km distance from the fault, owing to the inherent data noise exceeding the signal beyond this distance and introducing artifacts. InSAR data were included up to 100km distance owing to their lower intrinsic noise, to provide better constraints on the deeper part of the fault.”

2. Fault geometry:

The authors argue that no significant asymmetry is observed in the surface displacement field and therefore adopt a vertical fault geometry. However, looking at the azimuth and range offsets, it seems to me that the surface displacement field is slightly asymmetric and suggests an eastward dipping fault. Additionally, the aftershocks are predominantly located on the eastern side of the fault trace, which would also support this interpretation. I suggest that the authors consider inverting for fault geometry, as assuming an incorrect fault dip could bias the inferred slip distribution at depth.

We agree and thank all the reviewers for pointing this out; initially we had dismissed this asymmetry as residual data noise, but we have revised our model to include a dipping fault. Although the overall slip distribution is largely unchanged in this model (see figure S8 above), we do find that it fits the near-field data significantly better, and the identified dip angles agree remarkably well with previous estimates of the dip angle based on interseismic data. We have added the following:

New supplementary figure S7, showing the grid search results and reduced chi-squared values for varying dip angles. This is a relatively coarse grid search, but indicates a best-fitting dip of 65E in the northern part, and 80W in the southern part.

New supplementary figure S8, comparing the slip distribution for the dipping model (a) to the original, vertical model (b). Visually, the slip distribution is almost unchanged, and the magnitude and seismic moment are very similar, but the data fit is much better.

New supplementary figure S9, showing the map view and perspective view of the dipping fault mesh.

We also updated the text as follows:

Line 197: Revised this whole paragraph (previously discussing implications of a vertical fault) to read: “We constructed a finite slip model after carefully resampling the data to maintain accurate values close to the fault (see Methods). Prior interseismic observations near the Sagaing ridge had suggested variable eastward fault dip as shallow as 70 degrees^{22–24,42} and a possible westward dip to the south near Bago²². We conducted a grid search for two dip angles in the north and south of the rupture, with a smooth variation between them. The best-fitting model has an eastward dip of 65° in the north, and a westward dip of 80° in the south and fits the data significantly better than a vertical fault model, although the slip distributions are similar (Supplementary Figures S7-S9). The eastward dip in the north agrees well with the interseismic observations, and while significantly off-vertical, is not the most extreme example of a strongly dipping strike-slip fault⁴³.”

Line 517 (methods): added “We identified the best-fitting fault dip along the northern and southern parts of the rupture by conducting a grid search over two parameters for the dip angle in the northern and southern parts of the fault, with a linear variation in dip assumed across a 200km length centered at 20°N. The best-fitting model has a dip in the northern segment of 65°E, and 80°W in the southern segment (Figure S7), in surprisingly good agreement with previous interseismic geodetic inferences in the northern part^{22,23}. The slip distribution for both a vertical and dipping model is highly similar (Figure S8) but the data fit is significantly better for a dipping fault. The preferred fault geometry is shown in map view in Figure S9.”

3. Aftershocks distribution

L 96-98 and Figure 1. The statement that aftershocks are primarily located at the edges of the rupture should be somewhat tempered. As shown in Figure 1, the southern cluster of aftershocks extends along more than one-third of the rupture length, while the northern cluster appears is not strongly more concentrated than the distribution observed in the central section. Furthermore, it is unclear whether the apparent lack of aftershocks in the central part of the rupture is real or could be due to limitations in detection capability. Could the authors add the station locations on Figure 1?

Additionally, it is notable that all aftershocks are located on the eastern side of the fault. This spatial pattern deserves further discussion. Does it reflect fault geometry or other structural or detection-related factors?

Thanks for pointing this out, we agree this may have been too strongly phrased. We have moderated this sentence as follows: (**Line 120-124**)

“A large portion of the aftershocks were distributed between the epicenter and the rupture’s northern end, with fewer events along the southern 400 km of the rupture^{40,44} (Figure 1). This could be indicative of either a relatively simple fault without significant heterogeneities or off-fault planes of weakness, or a smooth supershear rupture in this segment with relatively uniform stress drop^{45,46}.”

Regarding the accuracy of these locations, we were also concerned about the eastward bias, and

made a comparison between the catalog from the U.S. Geological Survey and the TMD's catalog for both this earthquake and the 2018 Bago-Yoma earthquake (figures below). Our comparison suggests that even if we extended the catalog to 10 June based on the USGS's earthquake catalog, both catalogs show fewer earthquakes in the central and southern part of the rupture, but also that the TMD catalog's epicenters have a systematic shift when compared to the U.S. Geological Survey's global catalog and the relocated earthquake catalog from the local network.

This suggests that the spatial pattern of aftershocks in the TMD catalog was affected more strongly by the TMD's network geometry than the geometry of the Sagaing fault. To avoid further confusion, we decided to use the aftershocks from U.S. Geological Survey to replace the aftershocks from TMD network in the revised Figure 1.

[editorial note: third party material redacted]

Figure: comparison between USGS and TMD epicenters from March 1 to June 10, 2025. Two things are noted here: a general southward decrease, but an absence near the center of the rupture, and an eastward bias in the TMD locations relative to USGS. As a result of this observation, we have replaced our aftershock locations in Figure 1 with the USGS locations (red here, cyan in Figure 1).

[editorial note: third party material redacted]

Figure: comparison between USGS, TMD, and Fadil et al. (2021) relocations, showing a similar bias in the TMD locations.

Fadil, W., Lindsey, E. O., Wang, Y., Maung, P. M., Luo, H., Swe, T. L., et al. (2021). The January 11, 2018, Mw 6.0 Bago-Yoma, Myanmar earthquake: A shallow thrust event within the deforming Bago-Yoma Range. *Journal of Geophysical Research: Solid Earth*, 126, e2020JB021313. <https://doi.org/10.1029/2020JB021313>

Minor comments:

L 96-98 "indicative of a smooth rupture and relatively uniform stress drop." Please add references

We updated the sentence to “indicative of either a relatively simple fault without significant heterogeneities or off-fault planes of weakness, or a smooth supershear rupture with relatively uniform slip and stress drop^{42,43}.” (**line 122**)

This seems to be more accurate given the existing literature. The added references are:

42. Das, S. & Henry, C. Spatial relation between main earthquake slip and its aftershock distribution. *Rev. Geophys.* **41**, (2003).

43. Yabe, S. & Ide, S. Why Do Aftershocks Occur Within the Rupture Area of a Large Earthquake? *Geophys. Res. Lett.* **45**, 4780–4787 (2018).

L.111-112: I do not see any clear bend in the southern part of the rupture on Figure 2. It might be more appropriate to refer to Figure 1, where the bend near 18°N is indeed visible.

Fixed, thank you.

L 335: datasets  datasets

Fixed, thank you.

Figures:

Figure 1: It is difficult to distinguish the aftershocks from the background seismicity. I suggest using a distinctly different color for the aftershocks to improve visual clarity. Since the aftershocks all appear to be shallow (i.e., <10 km), it may not be necessary to color them by depth. Using a single color clearly different from those used for background events would make them stand out more effectively.

Thank you. We have modified Figure 1 accordingly, using a distinct color for the aftershocks (with updated locations as noted above). Please see the revised Figure 1.

Figure 2: For consistency, please also indicate the dates of the pre- and post-event Sentinel-2C images, as is done for the other datasets.

Added these dates, thank you.

Figure 3: While the meaning of the yellow star and red dashed lines can be easily inferred, the black solid and dashed lines are not explained and should be clarified. Additionally, the abbreviation "MDL" is not defined.

Thank you. We have modified Figure 3 and added this information in the figure and figure caption accordingly. Please see the revised Figure 3.

Figure 4: Caption for panel B is missing.

Thank you; we added a caption as follows:

“(b) shows a series of vertical slip profiles in blue within the indicated along-strike regions for each sub-panel. Profiles in red are from the zone reported to have experienced high-intensity ground shaking during the 1930 event⁴⁶.” (line 869)

Supplementary data:

I am not sure to understand Figure S1. It is described as showing the raw optical correlation results, but the data coverage seems to be different from that shown in Figure 1?

Thanks for pointing this out. The optical correlation showed in Figure S1 had been augmented with additional imagery to cover a larger area away from the fault, although these data were not used in the main text figures or modeling. The near-field ground deformation is nearly identical to our earlier processing result, shown in Figure 2 and Figure 3. To avoid confusion, we replaced the image correlation result in Figure S1 to the same version we used for the fault slip inversion and main text.

Reviewer #2 (Remarks to the Author):

This contribution is a timely analysis of the 2025 Sagaing earthquake focusing on InSAR and Sub-pixel optical image matching. As far as I can see no journal articles applying these techniques to this event have yet been published. The authors highlight the lack of a shallow-slip deficit and comment on the magnitude of slip in terms of the geodetic slip-accumulation rate and different models of fault slip-accumulation and release. The techniques used are appropriate and I believe the results are interesting and valuable. The interpretation is suitably cautious and potential inconsistencies are pointed out.

Thank you for the kind and helpful review!

The main weakness of the manuscript is the lack of constraints from seismology. I believe that there is an emerging consensus that this is a super-shear event, a fact which could have implications for their interpretation. I do not think that the lack of an original seismological analysis should prevent publication, but some comment on the possible role (or lack of role) of supershear rupture is probably necessary. The authors clearly considered this as there is some consideration in a figure, but this does not seem to have made it through to the text.

Thank you for the helpful comments. We have added some additional discussion regarding the effects of the supershear rupture; please see the more detailed comment below.

Some more comment on potential segment boundaries may also be useful and the authors may consider providing a little more analysis on the mismatch in dip between their model and the seismological results.

Related to the potential segment boundaries, we have added some text; please see our reply to the next point below.

Related to the fault dip, see our extended comments to reviewer 1 above. We have updated the model to include a dipping model that is very similar to the seismological results, which actually used dip values based on interseismic GNSS studies.

The text is very clear and well-written, but I have a few minor comments

Specific comments

72-74: I would normally expect 'segments' to mean distinct sections of a fault separated by some structural complication (e.g. a bend or step-over) which may therefore be expected to rupture separately over multiple earthquake cycles. Is that the case here? The detailed fault morphology (in this sense) is not presented very clearly. It wouldn't be surprising for a straight section of fault (a single segment) to occasionally fail in small events before a larger event ruptures its entire length. The Robinson et al. paper cited for the simplicity of the fault is also not very detailed. Is there a more focused study on the fault morphology?

Thanks for pointing this out. Prior work on segmentation of the Sagaing fault was based primarily on observations of small stepovers and other fault complexities that appeared to match well with historical earthquakes and historical earthquake intensity records. The most detailed studies were by Soe Thura Tun and Watkinson, 2017 (ref. 32) and Wang et al. 2014 (ref. 2); both of these papers provide some explanation for the segment definitions based on the surface fault geometry, but we agree the definitions are not as clear as on other faults worldwide. Except for a few fault splays near Naypyitaw, the section that ruptured in 2025 is very linear and continuous. To avoid confusion, we changed the word "segment" mentioned here to "sections", which hopefully implies a less strong geomorphic change along strike.

We have also further updated the sentence (**Line 83**) as follows, to be more descriptive about how the sections were defined: "Based on a combination of historical earthquake intensity records and along-strike changes in geomorphic expression of the fault, previous studies subdivided the central and southern Sagaing fault into five named sections^{32,38}, each of which is capable of generating M>7 earthquakes every ~100 to ~300 years^{2,27,39}, or rupturing together as in 2025."

83-84: What defines the ends of this section?

Thank you; related to the above. We added a small description here to clarify the ends of this particular section: "...the central section between the Sagaing ridge near Mandalay and a trans-tensional stepover near Naypyitaw..."

84-85: The cited paper is a bit more cautious in its assignment of the causative fault of the Ava earthquake than is implied here, but I agree that the 2025 event does make it likely that this interpretation is correct.

Thank you; we changed "believed" to "hypothesized" here to more accurately reflect the original interpretation.

94-96: The USGS finite fault model (version 3) states that the earthquake was supershear.

<https://earthquake.usgs.gov/earthquakes/eventpage/us7000pn9s/finite-fault>

Recent research publications confirm this (e.g Inoue, N., Yamaguchi, R., Yagi, Y., Okuwaki, R., Bogdan, E., & Tadapansawut, T. (2025). A multiple asymmetric bilateral rupture sequence derived from the peculiar tele-seismic P-waves of the 2025 Mandalay, Myanmar earthquake. *Seismica*, 4(1). <https://doi.org/10.26443/seismica.v4i1.1691>)

There does not appear to be much discussion of the potential role of supershear rupture other

than the separation of profiles in Fig 5a/b. I think there needs to be some treatment of this in the text.

Thanks for raising this issue. We know there are many groups working on the supershear aspect of this earthquake, and we have added some recent publications in the maintext, including the one above; the second half of this paragraph has been expanded to read:

(Line 118) “Recent seismological studies have confirmed the supershear nature of the rupture^{41–43}, in particular the southward-propagating portion of the rupture north of Naypyitaw. The northward rupture speed is slower, characterizing a subshear rupture between the earthquake epicenter and its northern termination. A large portion of the aftershocks were distributed between the epicenter and the rupture’s northern end, with fewer events along the southern 400 km of the rupture^{40,44} (Figure 1). This could be indicative of either a relatively simple fault without significant heterogeneities or off-fault planes of weakness, or a smooth supershear rupture in this segment with relatively uniform stress drop^{45,46}.”

Based on our analysis in Figure 5, we did not find that differences in the SSD are related to supershear vs. subshear rupture, though it’s true we did not really discuss this in the original text. We added some text as follows at **Line 294**:

“We find that rupture velocity (supershear or subshear) does not have a clear effect on the fault slip distribution with depth, as cases from both types of rupture showed various degrees of SSD (Figure 5a and 5b). This is supported by the rupture in the 2025 Mandalay earthquake, in which both the northward propagating subshear rupture, and the southward propagating supershear rupture^{41–43} show similar slip distributions with minimal SSD (Figure 4).”

119-120: Also state the length and limiting latitude to North.

Thank you; added “near 22.6°N”. **(Line 138)**

130-133: Was the effect of the fault dip assessed quantitatively or was the dip fixed at 90° due to lack of pronounced asymmetry. I would not expect a steeply dipping N-S pure strike-slip fault to be strongly asymmetrical (e.g. Sethanant, I., & Nissen, E. (2025). The InSAR lookbook: an illustrated guide to earthquake deformation interferograms. *Seismica*, 4(1). <https://doi.org/10.26443/seismica.v4i1.1413>) and given the fact that the seismological estimates do suggest a non-vertical dip it seems important to at least quantitatively test the geometry they prefer.

We fully agree, and have revised the model to have a non-vertical dip; see our detailed response to reviewer 1 above. This paragraph has been entirely rewritten in light of the new model.

156-159: This could be possibly be rewritten more clearly.

Thank you; we have revised these sentences as follows, and hope they are now more clear:

(Line 248) “The southern termination of the main rupture at a comparatively sharp bend in the

fault is also the location where the Dec-1930 earthquake was inferred to terminate, indicating a possible persistent rupture barrier. Finally, the base of the seismogenic zone aligns well with the depths of relocated background seismicity; in particular a small zone of high activity surrounding the 2025 epicenter, suggestive of a high pre-stress along that portion of the fault⁵³.”

187: Could you include a statement here about the horizontal resolution of your method (i.e. how distributed would OFD need to be to show up as such). This is in the methods, but I think it needs to be briefly stated here as well.

Agreed, we added this caveat to the end of the sentence: “except perhaps within the top ~100-200 m, the spatial resolution limit of our method^{49–53}”. **(Line 291)**

195: Also a supershear event?

Yes, both Avouac et al (2014) and Wang et al (2016) suggest that part of the 2013 Balochistan earthquake rupture was supershear, with rupture velocity reaching 3.7–4.1 km/s, close to or faster than the local shear wave velocity. Hence, in the Figure 5 we classified this event as a partially supershear rupture. We also changed our description to “...the 2013 Mw 7.7 Balochistan earthquake, a partially supershear event in Pakistan...” to make this more clear. **(Line 322)**

References:

Avouac, J. P., Ayoub, F., Wei, S., Ampuero, J. P., Meng, L., Leprince, S., ... & Helmberger, D. (2014). The 2013, Mw 7.7 Balochistan earthquake, energetic strike-slip reactivation of a thrust fault. *Earth and Planetary Science Letters*, 391, 128-134.

Wang, D., H. Kawakatsu, J. Mori, B. Ali, Z. Ren, and X. Shen (2016), Backprojection analyses from four regional arrays for rupture over a curved dipping fault: The Mw 7.7 24 September 2013 Pakistan earthquake, *J. Geophys. Res. Solid Earth*, 121, 1948–1961, doi:10.1002/2015JB012168.

201-202 Would you be able to resolve OFD at the lower end of this scale.

Likely not in the model, but the data have resolution at the scale of ~10s of meters, so it is close to resolvable. In fact, recent field work by several of us (WY and TZHT) suggests that this rupture does exhibit some off-fault deformation at the scale of tens of meters; this is also now more visible in the revised profiles in Figure 3. As discussed above and in the text, we draw a distinction between this type of near-field OFD that does not represent a reduction in total slip, and the geodetic SSD which does imply reduced slip near the surface and is the main subject of our analysis.

To more clearly address this in the main text, we added a short description at **line 331**: “Our analysis suggests that the 2025 Mandalay event also experienced some degree of OFD within tens of meters of the fault in some locations (Figure 3e). Both cases highlight that a minimal SSD does not necessarily preclude substantial energy dissipation via OFD at small scales surrounding the fault trace.”

208-209: I have some concerns about this statement because it is not that certain exactly what failed in this event, but it does seem likely the area that has the highest slip did so I suppose this comment is reasonable.

Thank you; we softened “expected” to “hypothesized” again here to acknowledge the uncertainty around the 1839 rupture.

347-350: Doesn't this belong in the results rather than the methods?

Agreed, thank you – we moved this sentence up to Line 210, and revised the numerical values to reflect the new model.

Reviewer #3 (Remarks to the Author):

The authors present an analysis of displacement data from the 2025 Mandalay, Myanmar earthquake with a goal of modeling the event and drawing inferences about fault behavior and properties. Using surface displacement estimates from optical image correlation, radar image correlation and InSAR, they produce a dislocation model of the slip in the earthquake, and highlight several atypical features of that model compared with other events from the past few decades: 1) along much of the fault, peak slip is at or close to the surface; 2) the fault rupture was unusually long for an earthquake of this size; 3) slip is quasi-slip predictable, in that areas of the with the longest time since the previous earthquake rupture slipped more than areas that had more recent ruptures; 4) the slip in some areas was greater than expected, given our estimates of fault slip rate and time since the last event.

The first of these findings leads the authors to make a broader point about the shallow slip deficit (SSD) for this event – or apparent lack of it. Compared with other events of a similar size and scale for which similar geodetic data are available, the Mandalay earthquake had significantly larger shallow slip, and a negligible SSD. In previous studies of SSDs of earthquakes, events that have large SSDs typically have surface ruptures that are distributed over wide fault zones (often hundreds of meters wide), with significant off-fault deformation accommodated by inelastic processes. In contrast, the authors suggest, the minimal SSD of the Mandalay event implies that the fault zone is highly localized and likely a result of a very mature fault, with minimal roughness and segmentation. Another interesting inference drawn from the model results include the possibility of some kind of 'dynamic overshoot' – a transient stress effect of the earthquake rupture – being a potential explanation for the slip in the earthquake exceeding the prediction based on slip rate and interval since the last earthquake.

Overall, this is a solid study that provides an interesting end-member data point in the literature about SSDs. The conclusions made are reasonable, and entirely consistent with the results presented. The text and figures are clear. I have a few questions about choices that were made in the modeling, but I do not think they will substantively change the findings of the study (but they might change some of the details), and I do not think they would require major corrections to address. I will start with my longest comments, and then move to line-by-line comments.

Thank you for the helpful review and positive comments!

1) Assumptions made in the dislocation model

One thing that I noted is that the authors' model has some baked-in assumptions that ought to be evaluated – that the fault dip is vertical, and that the rake is pure right-lateral. This differs from the USGS W-phase and Global CMT solutions, which could be considered 'average' solutions of the whole event, both of which suggest the fault dips to the east and has a minor dip-slip component of motion. One consequence of using this assumed geometry is that it could result in biases to the slip distribution – more slip could be required to match the observed surface deformation, for instance. This could be an explanation for the estimated moment magnitude from the authors' model exceeding seismic estimates by ~ 0.1 magnitude units (I cannot say exactly what the difference is in seismic moment, as I could not find that number in the manuscript – I would urge that such information be included). Indeed, I do not recall the authors commenting upon that difference.

Thank you – we have taken care to revise the model and address these assumptions in more detail: First, as noted above, we have revised the model to account for a non-vertical fault, with our revisions detailed in our response to Reviewer 1 (and 2). Suffice to say, the dipping model is much better and we thank all reviewers for encouraging us to undertake this modeling effort.

Second, with regard to the pure right-lateral rake: we also considered a model allowing dip-slip, as shown in the updated Supplementary Figure S11, reproduced here:

Supplementary Figure S11, showing the comparison between a strike-slip-only model (a) and a model allowing both strike- and dip-slip (b).

The overall pattern of slip along-strike and with depth is effectively unchanged, although it is somewhat smoother toward the south. We could potentially consider this our preferred model.

However, as noted in the methods (**Line 529**) we conducted an F-test to evaluate the relative likelihood of two models with differing numbers of parameters, and found that the two-slip-component model does not provide a sufficiently improved fit to justify the doubling in the number of parameters. This is reflected in the second model's higher $\chi^2/d.o.f.$ seen in the figure above, since the second model has 4,216 fewer degrees of freedom (the number of additional slip parameters in this model). This is of course only a limited justification of the choice to prefer the single-component model, but we hope it is sufficient in combination with the figure above showing that the results are nearly unchanged.

We also added a comment to note this decision more clearly in the main text (**Line 207**): “We also considered a model with two components of slip, but found that a model with strike-slip only is preferred by an F-test, although the two models are highly visually similar (Supplementary Figure S11).”

Finally, regarding the moment and magnitude – lack of a moment value was an oversight, thank you! We have added this on Line 211. Additionally, we noticed a rounding error in our original magnitude calculation (we used a constant offset value of 10.7, as opposed to the more accurate $(2/3)*16.1 = 10.7333\dots$). This change reduced our estimated magnitudes by 0.03, so it is now in slightly better agreement with but still somewhat higher (7.75) than the seismic observations (7.7).

We consider this to be within range of acceptability, particularly given uncertainty regarding the shear modulus (we assumed 30 GPa). This value has been found to provide relatively good agreement between seismic and geodetic moment in other earthquakes with typically deeper slip, but given the extremely shallow nature of slip in this event, this could be an overestimate.

We added a comment to this effect on **Line 529**: “We computed the seismic moment, magnitude and average stress drop for our models assuming a constant shear modulus of 30 GPa, but we note this could be an overestimate given the shallow nature of slip in this event. A value of 25 GPa would reduce the magnitude for our best fitting model from 7.75 to to 7.7, in closer agreement with seismic observations⁴⁰.”

It would be helpful for the authors to assess a few other fault geometries (e.g. the W-phase solution) to see what effect they have on the modeled slip, and also how well such models fit the data. I do not expect this would change the primary conclusion surrounding the SSD, as the estimated slip values from an alternate fault geometry are likely to vary in proportion to the values in the authors' current preferred model – the portions of the fault that slip the most will still be the portions that slip the most, shallow parts of the fault will still slip more than deeper parts, the SSD will still be negligible – but some of the secondary conclusions about the possible 'overshoot' of slip in the earthquake may be affected if the estimated slip changes (if it goes down, it could mean there is no overshoot... but if it goes up, then there could be more of an overshoot).

We agree that considering alternate fault geometries was a good idea; in lieu of directly implementing alternate reported geometries like the W-phase model, we hope that our above-mentioned grid search for dip values and additional model tests are sufficient evidence of the robustness of this event's slip pattern to variations in fault geometry and other parameter choices.

Related to the issue of slip 'overshoot', our dipping fault model has a nearly unchanged but slightly higher peak slip of 6.0 m, compared to 5.9 m in the vertical case. This is likely within the uncertainty in our model, and indicates a relatively low sensitivity of this value to the considered parameters.

We have noted the relative insensitivity of the model parameters in our discussion of the dipping model at **Line 202**: "The best-fitting model has an eastward dip of 65° in the north and a westward dip of 80° in the south, and fits the data significantly better than a vertical fault model, although the slip distributions, peak slip, and overall moment release are nearly unchanged."

2) How sensitive is the model to shallow slip?

If I read correctly, it seems that the fault geometry has a minimum element size of 1–2 km. I understand that this is a long rupture, and there are practical considerations about how small the elements can be for the inverse problem to be tractable. But one of the things I wonder is what the limits of the model could be for resolving shallow slip, and any deficits therein? What scale of SSD would be impossible to detect with the preferred fault geometry? I still think the shallow slip pattern is highly suggestive of the authors' conclusions, but it would be good to place limits on what can be said in that regard – perhaps with some kind of synthetic test?

We agree that there is a limit to our model's ability to resolve very shallow off-fault deformation (OFD) very close to the fault rupture, as discussed in more detail in response to Reviewer 2 above. Indeed, it is likely that there is some un-modeled OFD at a scale of tens of meters in some areas, as we have now noted in the main text (**Line 332**), but we draw a distinction between this near-field OFD, related to shallow inelastic effects in the soil, and the kilometer-scale geodetic SSD that is the main subject of our analysis.

3) Line-by-line comments

Line 64: Is Istanbul an appropriate comparison? How mature are the faults in the Sea of Marmara? My thought is that they are probably less mature (and straight) than the San Andreas. I also wonder where else in the world there are appropriately mature strike-slip faults near major cities? Guatemala City? Wellington, New Zealand? Any thoughts?

It is a fair point that the NAF near Istanbul is less mature and geometrically simple than the San Andreas, although it does have an inferred offset of ~50 km along the main trace according to Akbyaram et al. 2016, <https://doi.org/10.1016/j.tecto.2015.11.026>, which puts it closer to the "mature" side in our figure 5. We have added this as a reference in this sentence.

We also agree that mentioning other cities near mature strike-slip faults is a good idea. The Motagua fault in Guatemala appears to be similar to the NAF, or slightly lower (Guzmán-Speziale & Meneses-Rocha, 2000, [https://doi.org/10.1016/S0895-9811\(00\)00036-5](https://doi.org/10.1016/S0895-9811(00)00036-5)). The alpine fault is a good example with a total offset over 400km (Sutherland, 1999, <https://doi.org/10.1080/00288306.1999.9514846>), although the distance to Wellington means rupture directivity could play a major effect. Still, we added a mention to both faults in this sentence and included the above references.

The sentence now reads: “Similarities between the Sagaing fault, the San Andreas Fault in California, the Alpine fault in New Zealand³¹, the Motagua fault in Guatemala³², and the North Anatolian Fault in Turkey^{33,34}, among others, means the 2025 event provides a unique opportunity to study the rupture and deformation processes associated with large continental strike-slip earthquakes on mature faults, with implications for improved hazard estimates for nearby cities like San Francisco, Istanbul, and Guatemala City.” **(Line 60)**

Line 85: Approximately what was the "full slip" of the 1839 earthquake? It would be very interesting to know how it compares with the slip in this event, if that is at all possible?

We agree that this would be nice to know! Unfortunately, as noted by Reviewer 2, the true slip and extent of this event are still highly uncertain. The preliminary results in the following preprint: <https://eartharxiv.org/repository/view/9343/>, cited as Ref. 42, on which two of us are coauthors (WY and MT), identifies the likely trace of the 1839 rupture north of Mandalay. As described there, field studies conducted over the past decade estimated an average of 4-5 m of slip from the event prior to the 2025 earthquake, which is likely to have been the 1839 earthquake.

Given this uncertainty, we changed the phrase “fully slipped” to the more precise “slipped in a major event” here. **(Line 91)**

Line 129: It may be that changes in dip cannot be resolved, but I do not think that a consistent, non-vertical dip is ruled out by a lack of asymmetry.

We agree, and have described our updated dipping model in response to reviewer 1 above.

Lines 208–211: It occurs to me that if dynamic overshoot is a consistent feature of earthquakes on this portion of the Sagaing fault, then it should be included in geologic slip rates, but perhaps not in geodetic estimates of slip rate. It seems that all of the slip rate estimates quoted here are geodetic? If this reasoning is correct, then the difference between estimated and actual slip is in the direction you would expect, I think?

This is an intriguing idea, but we don't believe that dynamic overshoot could be a mechanism for producing consistent disagreement between geodetic and geologic slip rates – even if dynamic overshoot does occur regularly in large, shallow earthquakes like this one, then the deeper part of the fault would presumably catch up at some later time, either through accelerated afterslip or in smaller earthquakes, so that there should still be a net block offset with a constant slip rate at all depths within the fault zone, and geologic/geodetic rates would remain in general agreement.

In any case, there are a few geologic estimates of slip rate that are relatively consistent with the geodetic rates; owing to reference limitations we had omitted them, but we have now added them in the main text on line 338 as follows:

(ref. 26) Bertrand et al. (1998), 10-23 mm/yr spanning 250-300ka at 22.5°N, offset basalt flows
(ref. 27) Wang et al. (2011): 11-18 mm/yr spanning ~1-3ka at 17.5°N, offset fortress wall

There are a few other, older rates but they are much less certain. Overall, the combined 20-24 mm/yr rate we cite here is consistent with these values, and does not suggest a net upward bias in the geologic rates.

Lines 320–323: A single fixed rake that was not pure right-lateral would have the same number of degrees of freedom as the authors' preferred model though, so if it fit the data better, it would be significant. So it isn't "dip-slip" that has been ruled out, so much as "variable dip-slip".

This is true! However, our variable-rake model (supplementary figure S11) doesn't have an overall rake that consistently differs from pure strike-slip, rather it varies slightly along-dip and along-strike. If we did fix the rake at an average angle, it would be very close to pure right-lateral strike-slip: the area-weighted rake of the overall 2-component model is 182.6°, which we have added to the caption of figure S11. This matches well with field observations showing little or no detectable vertical motion across the rupture in most places. We have not done a search over all possible fixed rakes, but we could say that pure strike-slip is a good approximation to the best fixed-rake model.

Lines 332–333: My hunch is that 3 cm is pretty small as minimum errors in image matching go (it is low for InSAR too, but possibly defensible)? How did the authors arrive at that number?

This is certainly a low value, and is not representative of the average uncertainties! The resampled data uncertainties were determined for each data point by computing the standard deviation of all pixels within each quadtree resampling box, which gives an average uncertainty of 35-48 cm for the optical imagery and azimuth offsets, and 6-9cm for the range offsets, owing to the much higher range resolution of Sentinel-1. However, the distribution of uncertainties is quite wide and some pixels coincidentally had a very low uncertainty, which affected the model. All of the InSAR pixels also had very low uncertainties (less than 1cm) when using this method, which we considered unreasonable. To settle on 3cm, we gradually increased the minimum threshold until the model chi-squared value became close to 1.

There may be a more systematic way to handle these ultra-low uncertainty pixels, but we did not find it in the literature. Our idea comes from the GNSS literature where this “minimum uncertainty floor” is a common practice – e.g. Lindsey & Fialko, 2013 (<https://doi.org/10.1029/2012JB009358>) and many others. Hopefully it can be considered acceptable in this case too.

To better describe our process for deriving the uncertainties, we added some detail to the description in the methods section (**Line 555**):

“Data points were weighted inversely by the standard deviation of the original pixels within each downsampling quadtree cell, resulting in an average uncertainty of 36 cm for the optical data, 35-48 cm for the azimuth offsets, and 6-9 cm for the range offsets, owing to Sentinel-1’s higher range resolution. The InSAR phase uncertainties derived from this method were unreasonably small, less than 1cm, and a few pixels within the other datasets had similarly unreasonably small values. Therefore we imposed a minimum uncertainty floor of 3 cm for all datasets (10 cm for the track 106 phase, which was strongly affected by localized tropospheric disturbance near Mandalay) to prevent over-fitting of the InSAR and individual data points.”

Line 349: Please quote the actual seismic moment estimate for the model, either here or somewhere else in the manuscript.

Thanks for catching this oversight – this sentence has moved up to **Line 210** and the seismic moment is now noted there.

Figure 3b: Why not plot local fault-parallel offset here, rather than east and north separately? "conponent" in the plot legend is a misspelling

We have modified the figure and fixed the typo, and we now use the fault-parallel offset instead of the N- and E-component.

Figure 3c: Same comment as above about what is plotted here. Also, these profiles should be primary evidence for a lack of SSD. You would want to confirm that the peak offset is at the fault – is it? In Profile 21 in particular it looks like there could be some off-fault deformation, on the order of a few 100 m? (That would be in keeping with other studies where it has been identified, I think?) It is hard to tell exactly as the profiles are so zoomed out (each division on the scale is 2 km, usually SSDs are identified on a much shorter length scale than that.

Thanks for pointing this out. We checked our profile and found the apparent OFD is affected by the width that we used to plot these profiles. When we decrease the width of these profiles from 51 pixels (i.e., ~2 km) to 11 pixels (i.e., 440 m), this apparent OFD disappears. We have updated our Figure 3 to avoid this issue.

Reviewer #3 (Remarks on code availability):

I think the results would be potentially reproducible if the appropriate input files were provided, but there seems to be no link to the Zenodo repository in the manuscript document (although one is mentioned). The GitHub repository linked to does have test cases for the code.

Apologies, we had not yet put the files in the Zenodo repository, as we preferred to wait for the final version in case of major changes and to prevent confusion with multiple versions. We certainly will do this when the manuscript is in final form.

REVIEWERS' COMMENTS

Reviewer #1 (Remarks to the Author):

The authors have addressed all of my questions thoroughly and clearly. I believe the manuscript is now in excellent shape, and I am pleased to recommend it for publication in its current form.

Thank you for your comments and suggestions that helped us to improve this manuscript.

Reviewer #2 (Remarks to the Author):

The authors have addressed my earlier comments fully and I believe that manuscript has been significantly improved (especially by investigating the fault dip as requested by all reviewers). I believe that the manuscript is ready for publication.

Thank you for your comments and suggestions that helped us to improve this manuscript.

Reviewer #4 (Remarks to the Author):

This is a very interesting paper that explores the slip pattern in the 2025 Mandalay Earthquake. The authors use geodetic data to invert for the slip in this event, and find that it appears to have experienced peak slip right up at the free surface, with very little off-fault deformation (OFD). This result contrasts with many recent earthquakes, which display a shallow slip deficit and significant OFD. The authors attribute this result to the relatively straight, mature nature of this fault, and show that other recent earthquakes on faults with similar structures have displayed similar slip patterns. The results could have important implications for the near-fault deformation and damage, as well as radiated ground motion.

I found the paper straightforward to read, and their conclusions seem plausible. I don't have any major criticisms; my comments are largely editorial in nature. I think the paper will be publishable with minor revisions. I list below my line-by-line comments.

Thank you for your helpful comments and suggestions – we have made changes to clarify each of the suggested points as detailed below.

22: “Focused” is a rather vague term; could the authors use something more specific, or define how they are using this term?

Thank you for pointing this out. We modified this description to “...reveal a remarkably sharp surface rupture...” to make it more specific.

29, 72-73: Please clarify what is meant by “efficient” in this context. Many readers will assume it means radiation efficiency, but not having read the rest of the paper yet, it's not clear.

The meaning of using “efficient” here is to describe the concentration of slip on the fault in this event. We added a short explanation in the text to clarify the usage of “efficient slip” here: “slip concentrated on the fault with minimal off-fault inelastic deformation”.

129: I believe panel (d) is mislabeled as (e) in the caption to figure 2.

Thank you for pointing this out. We have corrected this figure caption.

204-205: This appears to be the definition of “efficiency” in this paper; it would be very good to provide it much earlier in the paper, the first time the term is used.

Agreed; we have added a brief explanation in the main text (72-73) to clarify this point earlier on, but left it here as well to maintain clarity.

218: Do you have a correlation coefficient or other numerical factor to quantify this inverse correlation? I can see it by eye in the diagram, but if it’s just a qualitative correlation, that should be made explicit.

We have modified our description to clarify that this is a qualitative inverse correlation based on the current dataset.

243: Hu et al. don’t question the presence of dynamic overshoot in strike-slip settings. There is no obvious reason why it wouldn’t occur there, as it is a physical manifestation of the momentum of the fault as it slides. In models it is typically a relatively small fraction of the entire slip, though.

Thank you for pointing this out. We have removed the phrase questioning the occurrence of dynamic overshoot on strike-slip faults to avoid this confusion.

254: I’d be wary about drawing too many conclusions from a single event. Figure 5 shows that other earthquakes, with similarly simple fault traces, may also have these properties. The conclusions would be strengthened if they mentioned these other corroborating events.

Thank you for pointing this out. We have extended the sentence and included the 1999 Izmit, the 2001 Kokoxili, and the 2002 Denali earthquakes, which shared some similarities to the 2025 earthquake in our conclusion sentence.